# Meta-unstable mRNAs in activated CD8+ T cells are defined by interlinked AU-rich elements and m6A mRNA methylation

Paulo A. Gameiro[1,2,3,12], Iosifina P. Foskolou [4,5,6,7,12], Yumna A. Butt [5,6], Aniek A. M. Martens [7,8], Klara Kuret Hodnik [9], Igor Ruiz de los Mozos [1,2], Nordin D. Zandhuis[5], Žan Hozjan[5,6,7], Veronica M. Kot [5], Rupert Faraway[1,2], Michiel Vermeulen [7,8,10], Monika C. Wolkers [5,6,7], Randall S. Johnson[4,11] ✉ & Jernej Ule [1,2,9] ✉

CD8+ T cells can rapidly produce effector molecules following activation. This activation triggers rapid changes in gene expression that rely on the control of mRNA levels via multiple mechanisms, including RNA modifications. N6-methyladenosine (m6A) is an abundant post-transcriptional modification that promotes the decay of messenger RNAs in the cytosol. However, how recognition of m6A sites is integrated with other regulatory mechanisms that alter the fate of immunoregulatory mRNAs in CD8+ T cells remains unexplored. Here, we apply the m6A-iCLIP and GLORI methods to identify the importance of m6A sites flanked by AU-rich elements (AREs) within the 3'UTRs of CD8+ T cell mRNAs. Presence of such ARE-flanking m6A motifs predicts meta-unstable mRNAs that rapidly decay upon CD8+ T cell activation. We demonstrate interdependent effects of mutations in the identified AREs and RRACHs on *TNF* mRNA stability. The ARE-flanking m6A sites in these mRNAs show particularly high iCLIP crosslinking of YTHDF proteins, which are also identified by proteomic interactome analyses along with additional novel RNA-binding proteins. Our study reveals a crosstalk between m6A and ARE-dependent mechanisms in CD8+ T cells, providing new approaches for modulating mRNA decay in T cell activation.

CD8+ T cells, also known as cytotoxic or killer T cells, play a crucial role in the immune response against malignant and virally infected cells[1]. When CD8+ T cells encounter infected or malignant cells, they proliferate and differentiate into effector cells (Teff), which produce high levels of effector molecules to kill target cells. After target cell clearance, most Teff cells undergo apoptosis, but importantly, a subset differentiates into memory cells (TM), which persist and provide long-term immunity[2]. When activated, CD8+ T cells undergo rapid changes

[1]The Francis Crick Institute, 1 Midland Road, NW1 1AT London, UK. [2]UCL Queen Square Institute of Neurology, Queen Square, WC1N 3BG, London, UK. [3]NOVA Medical School, Faculdade de Ciências Médicas, Universidade NOVA de Lisboa, Campo Mártires da Pátria 130, 1169-056, Lisboa, Portugal. [4]Department of Physiology, Development and Neuroscience, University of Cambridge, Downing Street, CB2 3EG Cambridge, UK. [5]MP Immunotherapy, Department of Research, Sanquin Blood Supply Foundation, Plesmanlaan 125, 1066 CX Amsterdam, The Netherlands. [6]Landsteiner Laboratory, Amsterdam UMC, University of Amsterdam, Meibergdreef 9, 1105 AZ Amsterdam, The Netherlands. [7]Oncode Institute, 3521 AL Utrecht, The Netherlands. [8]Department of Molecular Biology, Faculty of Science, Radboud Institute for Molecular Life Sciences, Radboud University, 6525 GA, Nijmegen, The Netherlands. [9]National Institute of Chemistry, Hajdrihova 19, 1001 Ljubljana, Slovenia. [10]Division of Molecular Genetics, The Netherlands Cancer Institute, Plesmanlaan 121, 1066 CX Amsterdam, The Netherlands. [11]Department of Cell and Molecular Biology (CMB), Karolinska Institutet, 171 65, Solna, Sweden. [12]These authors contributed equally: Paulo A. Gameiro, Iosifina P. Foskolou. ✉e-mail: randall.johnson@ki.se; jernej.ule@kcl.ac.uk

in gene expression that rely on finely-tuned control of messenger RNA (mRNA) levels[3]. An important determinant of mRNA stability in CD8[+] T cells are the adenylate-uridylate-rich elements (AU-rich elements; AREs)[4], abundant sequences in the 3′ untranslated region (3′UTR) of many mRNAs; these are particularly prominent in cytokine mRNAs[5,6]. AREs recruit RNA-binding proteins (RBPs) that promote either RNA stabilisation (e.g., HuR) or degradation (e.g., ZFP36L1/L2), thus representing an important mechanism of regulation of effector molecule production by $T_M$ cells after T-cell activation[3,7].

Another determinant of cellular mRNA stability is *N*6-methyladenosine (m6A) methylation. The m6A modification is co-transcriptionally deposited on mRNAs by the core methyltransferase complex METTL3/METTL14/WTAP[8], which plays widespread regulatory roles in mRNA metabolism; these include nuclear export, splicing, stability, translation, and compartmentalisation[8–12]. m6A demethylation is catalysed by α-ketoglutarate (αKG)-dependent dioxygenases, which include FTO and ALKBH5 demethylases[13,14]. Deposition of m6A is largely associated with accelerated mRNA metabolism through the activity of m6A reader proteins, including YTHDF proteins (YTHDF1, YTHDF2 and YTHDF3), that promote mRNA decay in the cytoplasm and P-bodies[15,16]. Recognition of m6A-modified RNAs is crucial in cell differentiation and development[17,18], and dysregulation of m6A deposition has been reported in leukaemia, aging and neurodegeneration[19–21].

Various next-generation sequencing (NGS) methods enable the transcriptomic profiling of individual m6A modifications by applying chemical, enzymatic, or antibody-based recognition of modified ribonucleosides and/or direct RNA sequencing to detect m6A residues on mRNAs[22–24]. One of these methods for profiling m6A at single-nucleotide resolution is miCLIP (also known as m6A-CLIP)[25,26]. This method relies on the use of anti-m6A antibodies for the immunoprecipitation of m6A-modified RNA fragments, which form covalent crosslinks with the antibody upon exposure to UV-C light. Insights from miCLIP and other methods demonstrate that m6A is primarily present in consensus RRACH (R, purine; H, not G) sequences, with greatest enrichment near stop codons at the start of 3′ untranslated regions (3′UTRs) of mRNAs[25–28]. Several methods have also reported signals in non-RRACH motifs, with variation between cell types and methodologies used. For instance, non-RRACH pentamers such as GGAUU were reported by the miCLIP2 study in HEK293T and C643 lines[29]. Direct RNA sequencing revealed 187 m6A sites in AGAUU in Arabidopsis mRNAs[30]. However, the roles of such non-RRACH sites in the m6A-dependent regulation of mRNA metabolism have not been systematically investigated.

Evidence shows that m6A modification is an important factor in immune-modulatory responses, including antitumor immunity[31]. Deletion of the reader protein YTHDF1 in mouse dendritic cells enhances antitumor responses through increased tumour antigen presentation to CD8[+] T cells[32]. Deletion of METTL3 blocks mouse CD4[+] T cell differentiation by keeping the cells in a naïve state[33]. However, we do not yet fully understand how the recognition of m6A sites is integrated with other regulatory mechanisms to control mRNA stability during T cell activation. It is clear that diverse RNA-binding proteins (RBPs), in addition to canonical YTH domain m6A readers, can regulate mRNA stability in m6A-dependent ways[9,16]. For instance, m6A mRNA methylation can either promote or inhibit the binding of HuR to transcripts such as *SOX2*[34] and *FOXM1*[35], or to various regions in a single mRNA[18]. It is thus important to characterise how m6A deposition might interact with various regulatory motifs to control the fate of immunoregulatory mRNAs in T cells.

Here, we used an improved iCLIP protocol (iiCLIP)[36] in the context of miCLIP to profile m6A in human CD8[+] T cells before and after activation. As expected, the strongest enrichment of the miCLIP signal was found in RRACH motifs (miCLIP-RRACH), but we have also identified enrichment within ARE motifs (miCLIP-ARE). Absolute quantification

of m6A sites using the antibody-independent GLORI method[37] revealed that miCLIP-ARE sites contain RRACH motifs within ± 4 nucleotides of AREs, which may represent a distinct functional subclass of m6A sites.

Through measurements of mRNA stability, we showed that mRNAs in which we identified ARE-flanking m6A sites are meta-unstable, so that they become rapidly degraded in activated CD8[+] T cells. Interestingly, these mRNAs encode late-effector and memory proteins. Through supervised learning of mRNA half-life data, we found that the combined miCLIP and GLORI signal in ARE-flanking m6A sites is more effective at predicting meta-instability than either signal in canonical RRACH motifs alone. We investigated these relationships further by manipulating the m6A machinery and with reporter assays. We found that METTL3 inhibition increases the stability of target mRNAs harbouring ARE-flanking m6A sites, such as *IL7R* and *TNF*. We also demonstrated that the impact of mutations in miCLIP-ARE sites on the stability of the 3′UTR-*TNF* mRNA depends on mutations in RRACH sites. Finally, we showed that miCLIP-ARE sites were enriched in iCLIP crosslink sites of YTHDF proteins and newly identified ARE-bound RBPs uncovered in this study, particularly members of the LSM family. These findings indicate that canonical m6A-modified RRACH motifs and the ARE-flanking m6A sequences represent functionally distinct m6A sites, bound by different m6A-reader RBPs. Together, they likely coordinate the meta-decay of mRNAs involved in key regulatory functions of CD8[+] T cells.

## Results
### ARE-flanking m6A sites revealed by motif integration of miCLIP and GLORI profiling

We applied a methylation-improved iCLIP (miCLIP)[25,26,36] assay to study m6A methylation in non-activated and activated CD8[+] T cells, where an anti-m6A antibody was used to immunoprecipitate fragmented RNA isolated from primary human CD8[+] T cells subjected to in vitro activation with aCD3/aCD28 beads (Fig. 1A and Supplementary Fig. S1A). Our miCLIP workflow introduces several features to identify motifs at anti-m6A crosslinks with high specificity: (a) m6A immunoprecipitation with or without UV irradiation of the m6A antibody to RNA to account for the technical impact of UV crosslinking[38]; (b) sequencing of fragmented input RNAs with the same protocol, to account for the technical biases of cDNA ligation and other aspects of the protocol[39] and allow us to normalise data by RNA abundance[40,41]; (c) use of UMIs and the iCLIP analysis approach to identify unique cDNA molecules; and (d) use of positionally enriched k-mer analysis (PEKA)[42] to identify the motifs enriched around high-confidence (thresholded) crosslinks, i.e., crosslinks located within peaks of high crosslink density, relative to crosslinks located outside peak regions; peaks were determined using either Clippy or Paraclu[43] (Methods) (Fig. 1A). Hierarchical clustering of 5-mers based on PEKA scores of thresholded sites revealed enrichment of crosslinks in GGACU and other variants of the RRACH consensus as well as motifs corresponding to the WWAWW consensus (W = A or U), reminiscent of the AU-rich elements (AREs) (Fig. 1B). The two sets of motifs were enriched at the starts of miCLIP reads for non-activated (noAct), Day1- and Day5-activated CD8[+] T cell samples but not at the starts of cDNAs prepared from input RNA (Fig. 1B). Noticeably, both the RRACH and WWAWW consensus sequences were enriched at the start of miCLIP reads in the absence of UV crosslinking (Fig. 1B), although some W-rich sequences were UV-dependent (Fig. 1B). This finding suggested that UV irradiation only partially promotes the crosslinking of nearby uridines to the anti-m6A antibody.

The metagene distribution of motifs around the starts of miCLIP cDNAs demonstrated direct alignment with the A motif corresponding to GGACU (and RRACH consensus) and proximal to UUAUU (and WWAWW consensus), but not those corresponding to CRACH consensus (e.g., CGACU) (Supplementary Fig. S1B), which is in agreement with reports showing that METTL3 does not exhibit in vitro activity on CGACU-containing RNA probes[44]. The starts of iCLIP reads can be

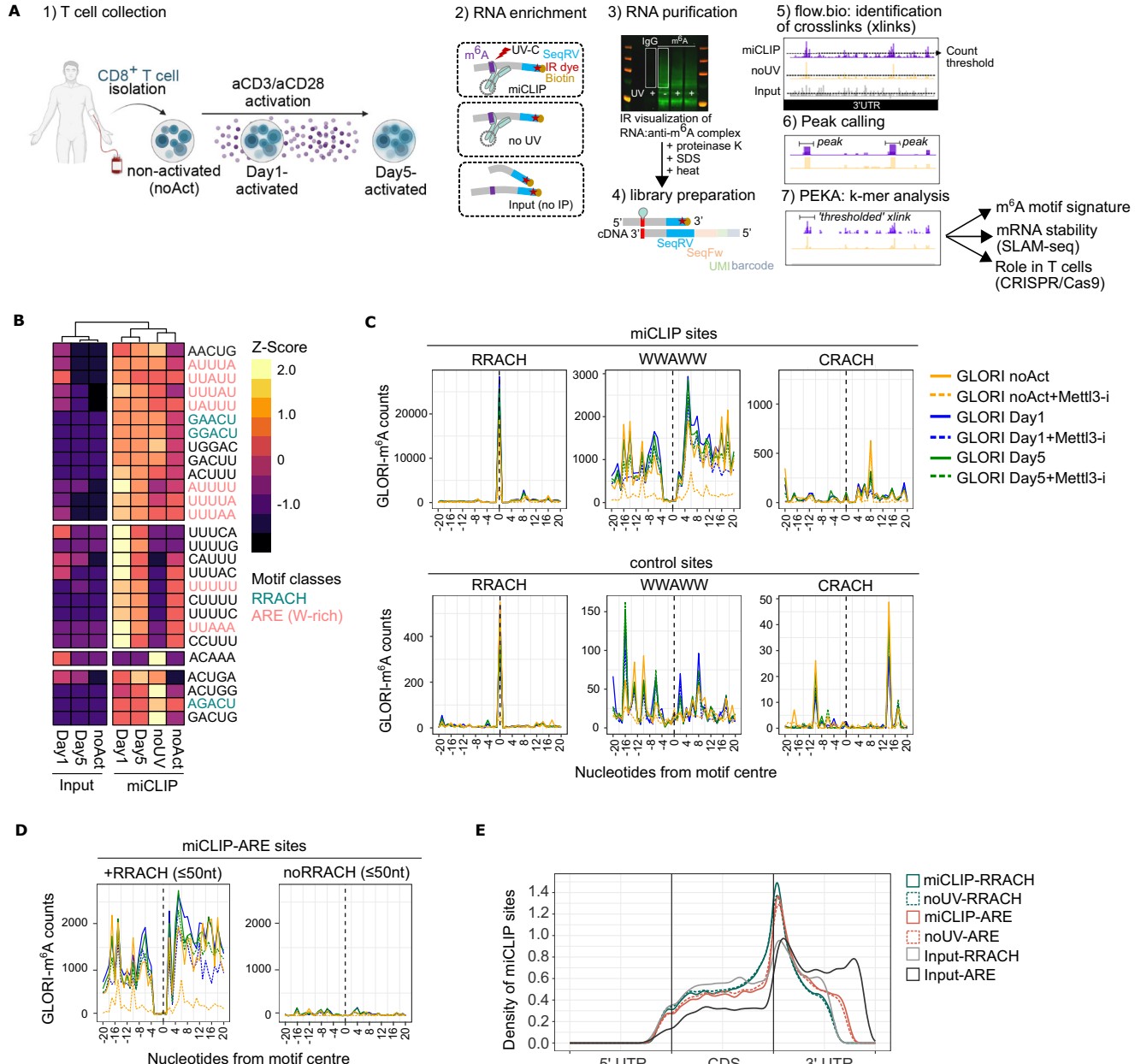

**Fig. 1 | Motif-specific profiles of miCLIP crosslinks in human CD8⁺ T cells. A** The miCLIP workflow for identification of high-confidence crosslinks: (1) primary human CD8⁺ T cells subjected to in vitro activation with aCD3/aCD28 beads; (2) RNA enrichment and experimental controls, including m⁶A immunoprecipitation without UV irradiation (noUV) and input RNA samples; (3) SDS–PAGE purification of RNA fragments crosslinked to the anti-m⁶A antibody followed by RNA isolation; (4) library preparation; (5) iCLIP analysis for determination of uniquely mapped crosslinks; (6) determination of peaks, i.e., regions with high density of crosslinks; (7) 'positionally enriched k-mer analysis' (PEKA) to extract sequences in high-confidence ('thresholded') miCLIP crosslinks. **B** Hierarchical clustering of 5-mers based on standardised PEKA scores at 'thresholded sites'; here, crosslinks and Clippy peaks were used for PEKA. The median PEKA score of each miCLIP or Input group (noAct, Day1 or Day5) was calculated for all 5-mers, and the unique sets of the 20 most enriched and variant 5-mers were plotted on the heatmap. **C** Metagene distribution of GLORI-determined m⁶A counts at miCLIP sites (top) and number-matched control sites randomly selected from the 3'UTR (bottom), centred on RRACH, WWAWW and CRACH pentamers; GLORI counts show the effect of the METTL3 inhibitor at each position, and represent the average across two CD8⁺ T cell replicates. **D** Metagene distribution of GLORI-determined m⁶A counts centred on miCLIP-ARE sites that contain nearby RRACH motifs (left) or lack RRACH motifs (right), showing the effect of the METTL3 inhibitor. Stratification by distance highlights the spatial relationship between m⁶A deposition and RRACH proximity. The WWAWW + RRACH sites refer to miCLIP-ARE sites containing at least one RRACH motif within ≤ 50nt. **E** Density plot showing the motif-specific distribution of PEKA crosslinks in the miCLIP and Input samples around scaled mRNA regions. For all panels, CD8⁺ T cells from healthy donors were isolated; not activated (noAct; *n* = 4), Day1 after activation (Day1; *n* = 5), Day5 after activation (Day5; *n* = 4), Mock (*n* = 12), noUV (*n* = 3). Panel (**A**) was created in BioRender. Foskolou, I. (2026) https://BioRender.com/ h18sinc. Source data are provided as a Source Data file.

enriched at uridines due to nucleotide biases of UV crosslinking, but the starts of miCLIP reads were aligned within two nucleotides from A, both at the RRACH and WWAWW motif centres (Fig. 1B and Supplementary Fig. S1B). On one hand, the UV independence of motif

enrichment, and alignment of cDNA starts with As, indicates that the RNA-bound part of the antibody is carried over throughout the miCLIP procedure and can induce cDNA truncation at its RNA contact sites even when it is not UV crosslinked. This is surprising, since the

procedure involves high-salt washing of the IP mixture, separation of the antibody-RNA complex via SDS–PAGE, transfer to a membrane and treatment with proteinase K to release an antibody peptide bound to RNA. Accordingly, we detected an infrared signal on the membrane in all the miCLIP and noUV samples that were immunoprecipitated with the anti-m6A antibody but not in the UV-crosslinked samples immunoprecipitated with a control IgG isotype (Supplementary Fig. S1C). Notably, the m6A antibody was developed against a monomer of *N*6-methyladenosine conjugated to a keyhole limpet haemocyanin lacking any WWAWW motifs; therefore, we speculated that contacts with AREs might be mediated by the presence of m6A on these sites.

Typical iCLIP relies on the analysis of cDNA starts as signatures of cDNA truncations induced by the crosslinked peptide[45,46]. Moreover, less frequent readthrough of reverse transcriptase (RT) across crosslinked peptides can lead to deletions and insertions, which can be analysed for cross-linking-induced mutation sites (CIMS)[25,47]. Expectedly, the incidence of deletions was higher in miCLIP than in input samples (Supplementary Fig. S1D). In this regard, it is documented that m6A-modified RRACHs result in CIMS characterised by C > T substitutions[25]. To assess motif-specific CIMS, we expanded the CIMS analysis to include C > T, T > A and A > T transitions to account for transitions in the ARE sites. While C > T substitutions were enriched at the genomic coordinates of RRACHs, T > A substitutions were enriched within AREs, but not detected at CRACH centres (Supplementary Fig. S1E). These substitutions were enriched regardless of whether UV was used in miCLIP. This indicates that the bound antibody peptide might be able to maintain bound to the RNA fragments throughout the CLIP procedure and form direct contacts with RRACHs and AREs, so that it can affect the mutation and truncation rate of reverse transcription at these motifs.

To assess the signal in AREs in other datasets, we analysed the published miCLIP2 data, where 661 m6A sites were reported in non-DRACH sequences[29]. By assessing the distribution of PEKA crosslinks around relevant motifs, we found significant enrichment of miCLIP2 crosslinks centred at GGACU and UUAUU pentamers, but not CGACU, in miCLIP2 data from both mouse embryonic stem cells (mESCs) and human cell lines (HEK293T and C643) (Supplementary Fig. S1F, S1G). In mESCs, the PEKA crosslinks were Mettl3 dependent at GGACU centres, but the signal at UUAUU centres was less affected by Mettl3 loss (Supplementary Fig. S1F). Antibody-independent methods also enable m6A profiling with single-base resolution, although some are experimentally tailored to specifically probe RRACH or related subset sequences, such as MAZTER-seq[48] and m6A-REF-seq[49] that rely on endoribonuclease recognition of ACA sequences. Among these methods, the recently developed GLORI features glyoxal- and nitrite-mediated deamination of unmodified adenosines, enabling the absolute quantification of m6A sites with high specificity[37]. By analysing GLORI-processed data in HEK293T cells, we observed that m6A peaks were enriched at RRACH and depleted directly at WWAWW pentamers enriched in their vicinity (Supplementary Fig. S1H). We further interrogated m6A sites previously identified using the antibody-free MePMe-seq method in HeLa cells[50]; these sites were also mildly enriched within 10 nucleotides from WWAWW pentamers but depleted directly at the 'A' centre (Supplementary Fig. S1I). Conversely, both antibody-independent methods showed that m6A sites were depleted at the 'A' centre yet not enriched in the vicinity of CRACH pentamers, with fivefold lower counts at each position when compared to WWAWW motifs (Supplementary Fig. S1H, S1I). Thus, our miCLIP data and that of Kortel et al.[29] revealed WWAWW-centred (ARE-like) motifs, which we denoted miCLIP-ARE sites. Conversely, two antibody-free methods in immortalised cell lines showed that m6A sites are enriched in RRACH motifs and proximal to WWAWW motifs, yet depleted at the WWAWW-centred A (Supplementary Fig. S1H, S1I). This led us to hypothesise that miCLIP-ARE sites may represent canonical RRACH motifs flanked by AREs in their immediate vicinity. To test this in a

METTL3-dependent manner, we performed GLORI in primary CD8+ T cells under activation conditions, with or without a competitive METTL3 inhibitor[51]. GLORI-detected m6A sites (GLORI-m6A counts) were highly enriched in miCLIP-RRACH sites (Fig. 1C). Notably, a substantial fraction (10%) of GLORI-m6A counts localised within ± 4 nucleotides of miCLIP-ARE sites, with signal peaking just 2 nucleotides downstream, while remaining low at the central A of WWAWW motifs (Fig. 1C). These signals were METTL3-dependent and most reduced in the non-activated T cells, suggesting functional roles (Fig. 1C – orange line). Importantly, the GLORI-m6A sites were more enriched around miCLIP-ARE sites (i.e., 8056 thresholded sites) than around randomly sampled 3'UTR ARE sites (i.e., 8056 control sites). Conversely, the GLORI signals at miCLIP-RRACH sites showed a similar enrichment pattern to those at randomly sampled RRACH sites, albeit with higher overall signal intensity (Fig. 1C). The data strongly suggest that miCLIP-ARE sites correspond to canonical RRACH motifs with flanking AREs within (±) 4 nucleotides of the 'A' centre.

To further dissect this motif relationship, we stratified miCLIP-ARE sites based on the proximity of RRACH motifs within ± 5, 25 and 50 nucleotides. In agreement with the preceding meta-analyses, the GLORI-m6A counts were enriched at miCLIP-ARE sites with flanking RRACHs, but not at miCLIP-ARE sites lacking nearby RRACH motifs (Fig. 1D and Supplementary Fig. S1J). Together, these findings uncover a class of m6A sites corresponding to RRACH motifs flanked by AREs – termed ARE-flanking m6A sites. Finally, consistent with previous reports, the overall miCLIP signal was most enriched around stop codons (Fig. 1E). Moreover, miCLIP-ARE sites displayed specific enrichment at the start of 3'UTRs, which contrasted the input signal in AREs, which was more evenly distributed across the entire 3'UTR (Fig. 1E). Importantly, the omission of UV crosslinking from miCLIP (noUV) had very little effect on motif distribution, consistent with antibody-RNA complex detection by SDS-PAGE. Thus, ARE-flanking m6A sites (miCLIP-AREs) may represent a distinct functional subclass when compared to canonical miCLIP-RRACH sites.

## Meta-instability of 3'UTRs with ARE-flanking m6A sites upon T cell activation

The m6A modification promotes mRNA decay in various cell types, but its effects on individual mRNAs vary significantly[9]. In this respect, which m6A sites play a role in fine-tuning transcript levels for dynamic and stimulus-dependent responses is not well understood. To assess mRNA-specific features in CD8+ T cells, we partitioned the miCLIP and input signal into RRACH and ARE motifs, normalised the counts using the DESeq2 model, and compared the hierarchical clustering of 3'UTRs across miCLIP and input samples. First, principal component analysis (PCA) of miCLIP counts across the top 1000 most variable 3'UTRs showed that miCLIP-RRACH and miCLIP-ARE motifs were clearly separated along the first two principal components (Supplementary Fig. S2A). Second, we observed that the miCLIP counts were poorly correlated with input counts in the respective motif (Supplementary Fig. S2B – light squares), indicating that changes in the miCLIP signal do not arise from changes in mRNA expression or stability[3]. Hierarchical clustering of mRNA 3'UTRs revealed 4 clusters characterised by different miCLIP signatures across different CD8+ T cell states: 3'UTRs with high miCLIP-ARE signal and low miCLIP-RRACH signal (cluster 1), lowly expressed 3'UTRs with high miCLIP-ARE and miCLIP-RRACH signal (cluster 2), 3'UTRs with low miCLIP-ARE and high miCLIP-RRACH signal (cluster 3), and highly expressed 3'UTRs with varied degrees of miCLIP signal in AREs or RRACHs (cluster 4) (Fig. 2A). Cluster 4, defined by highly expressed mRNAs mainly in activated CD8+ T cells (Fig. 2A), contained mRNAs typically upregulated during T cell activation; these included genes involved in macromolecule biosynthetic processes (e.g., *RPL15, RPL23, RPL31, RPL38, EIF4G1, MRPL3*), MHC II class (*HLA-A*), glycolysis (*PGK1*), lactate dehydrogenase (*LDHA* and *LDHB*), and granzyme B (*GZMB*) (Supplementary Fig. S2C and

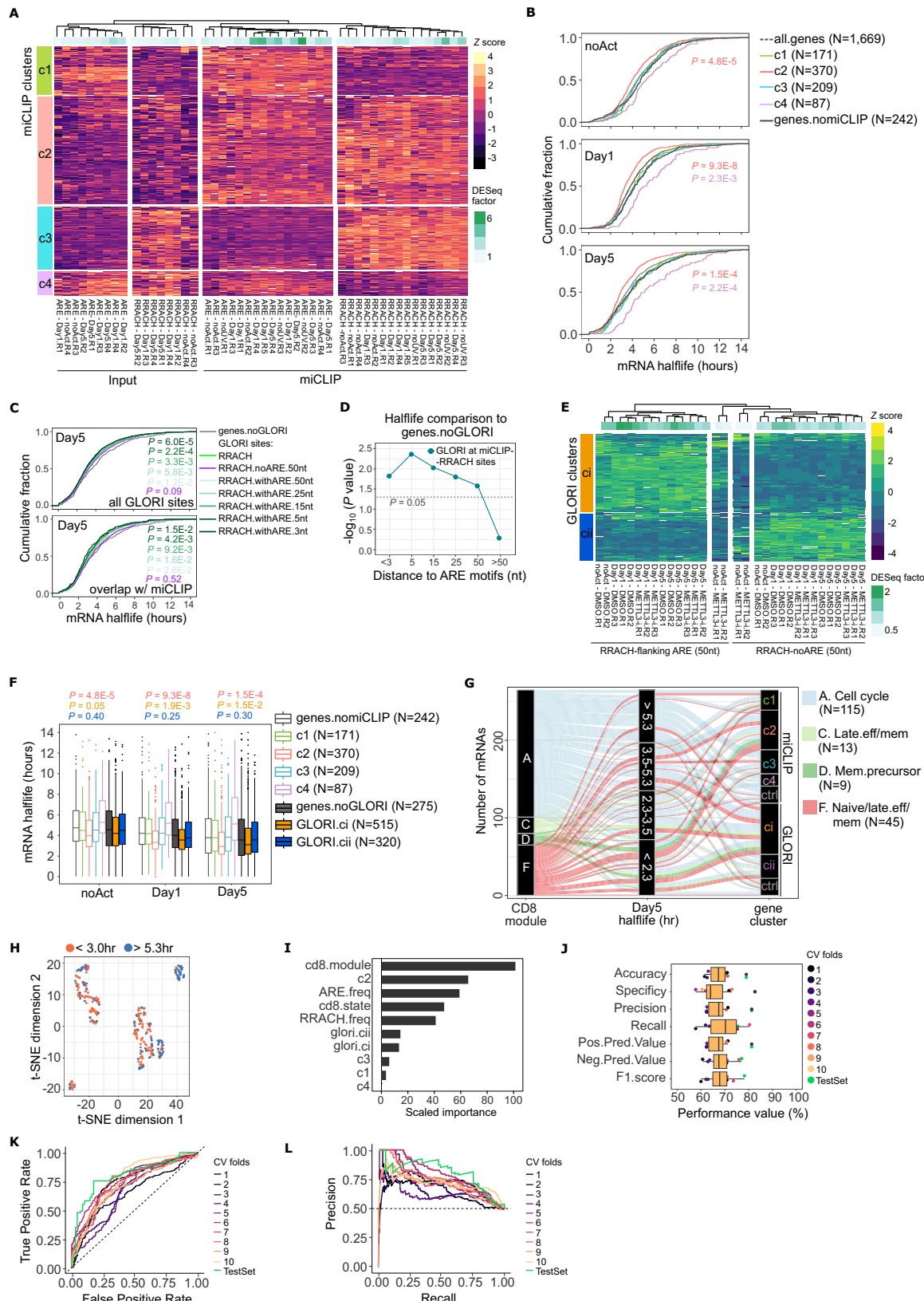

Supplementary Data 1). In contrast, cluster 2 included mRNAs involved in immunoregulation and mRNA metabolism. For example, cluster 2 contained mRNAs that play a crucial role in regulating cytokine production (*ZFP36L1, ZFP36L2*), immune activation (*CD28, CD69, TNFAIP3, GBP1*), and T cell memory (*BCL2, CXCR4, IL7R, TCF7*), TNF signalling genes (*TNF, TNFRSF1B, TNFAIP8, TNFRSF9*), other interleukins and

cytokine-related genes (e.g., *IL7R, IL10RA, IL27RA, CXCR6, CCR4, CCL3*), RNA splicing (*SRSF4*), and relevant RNA processing (*LSM1*) and stress granule assembly factors (*G3BP2*) (Supplementary Fig. S2C and Supplementary Data 1)[3]. Conversely, cluster 3 included mRNAs that regulate ribosomal processing and biogenesis (*DDX54, DEDD2, EIF6, RRP36*), T cell activation and innate immunity response (e.g., CXCR3,

**Fig. 2 | Dynamics and RNA stability associated with miCLIP-ARE and miCLIP-RRACH sites in CD8+ T cells. A** Heatmap of miCLIP (PEKA crosslinks) and input counts in 3′UTRs, normalised using DESeq2 size factors. The top 1000 most variable 3′UTRs across noAct ($n = 4$), Day1 ($n = 5$), Day5 ($n = 4$), and input ($n = 12$) samples are shown, clustered into four groups (c1−c4) defined by unsupervised learning. **B** Cumulative distributions of mRNA half-lives for miCLIP clusters (c1−c4) and transcripts without miCLIP crosslinks (genes.nomiCLIP), across noAct, Day1 and Day5 CD8+ T cells. Two two-sided Kolmogorov−Smirnov (KS) tests were used for statistical significance. **C** Cumulative distributions of mRNA half-lives for transcripts containing GLORI-identified m6A sites at RRACH motifs, stratified by distance to adjacent ARE (WWAWW) motifs. Top: all RRACH sites; Bottom: RRACH sites overlapping miCLIP-ARE peaks. Two-sided KS tests were applied. **D** KS test statistics comparing mRNA half-lives between transcripts lacking GLORI signal and those harbouring GLORI m6A sites at RRACH motifs, binned by distance to AREs. **E** Heatmap of GLORI m6A counts in 3′UTRs, normalised by DESeq2, for the 1000

most variable transcripts across RRACH-only and RRACH-flanking-ARE sites. GLORI clusters (ci, cii) were defined by unsupervised clustering. **F** Box plots of mRNA half-lives for miCLIP clusters, GLORI clusters, and corresponding no-signal controls across noAct, Day1 and Day5. Boxes represent the interquartile range (IQR), centre lines show medians, whiskers extend to 1.5 × IQR, and minima/maxima are shown as individual points. Two-sided KS tests were used for pairwise comparisons. **G** Sankey diagram showing the flow between CD8+ gene modules, Day5 mRNA half-life categories, and miCLIP/GLORI clusters. **H** t-SNE projection of transcripts grouped into short (< 3 h) and long (> 5.3 h) half-life classes using miCLIP and genomic features. **I** Variable importance from 30 conditional Random Forest (cRF) models predicting mRNA stability classes using miCLIP/GLORI cluster identity, genomic features and CD8+ T cell annotations. **J** Performance metrics of the final cRF classifier and 10-fold cross-validation. Box plots defined as in (**F**). **K, L** ROC (**K**) and precision−recall (**L**) curves for test data and each cross-validation fold. N denotes the number of mRNAs tested in (**B**), (**F**) and (**G**).

*GZMH, CD74, CD7, CD44*) (Supplementary Fig. S2C and Supplementary Data 1).

To assess the effect of CD8+ T cell activation on mRNA stability and its links to dynamic m6A sites, we performed SLAM-seq (thiol(SH)-linked alkylation for metabolic sequencing of RNA)[52] and determined mRNA half-lives in noAct, Day1 and Day5 activated CD8+ T cells. We performed a pulse-chase design of 24 h labelling with 4-thiouridine (4SU) pulsed every 4 h, followed by washout with uridine for 0, 0.5, 1, 3, 6 and 16 h. By fitting the T > C transitions over time to an exponential decay model and applying a goodness of fit cutoff ($R^2 > 0.5$), we determined reliable half-life estimates ($t_{1/2} = \ln(2) / k$, $k$ = decay constant) for 4158, 5585 and 5866 transcripts in noAct, Day1 and Day5 activated CD8+ T cells, respectively (Supplementary Fig. S2D and Supplementary Data 1). The median mRNA half-life was reduced in activated CD8+ T cells, from 3.6 h in noAct to 3.2 h in Day1 and 2.2 h in Day5 activated T cells (Supplementary Fig. S2D). The mRNA half-lives determined via SLAM-Seq agree with previously published mRNA half-lives in other mammalian cells[52–54] and with the expected high mRNA turnover in activated T cells[55]. Comparing mRNA half-lives between the miCLIP clusters revealed that mRNAs with miCLIP-ARE and miCLIP-RRACH sites (cluster 2) had shorter half-lives when compared to mRNAs with low miCLIP signal (i.e., nomiCLIP, ≤ 2 miCLIP counts averaged across samples) included in the DESeq2 model (Fig. 2B). Conversely, highly expressed mRNAs with low miCLIP-RRACH signal (cluster 4) had the longest half-lives (Fig. 2B). We next examined mRNA half-life distributions in relation to GLORI-determined m6A sites (GLORI sites), stratified by the proximity of ARE motifs (WWAWW) located within ± 3, 5, 15, 25 or 50 nucleotides of a RRACH motif. Consistent with our miCLIP data, RRACH-flanking ARE sites were associated with significantly shorter mRNA half-lives compared to RRACH sites lacking nearby AREs (i.e., no WWAWW motifs within ± 50 nt) (Fig. 2C). Notably, in Day 5 activated T cells, the mRNA half-lives progressively decreased as the distance between RRACH and ARE motifs narrowed (Fig. 2C, D). The significance threshold ($P = 0.05$) separated clearly the RRACH-flanking versus RRACH-lacking ARE sites within ±50 nucleotides (Fig. 2D). Relative to genes with low miCLIP (nomiCLIP) or low GLORI (noGLORI) signal, the miCLIP-cluster 2 and GLORI-cluster i were the most destabilised in Day5 activated T cells (Fig. 2C), even though all mRNAs were overall destabilised following CD8+ T cell activation (Supplementary Fig. S2E). Therefore, the datasets support the notion that activation of CD8+ T cells leads to faster decay of mRNAs that harbour ARE-flanking m6A sites (miCLIP-cluster 2 and GLORI-cluster i).

To test whether GLORI-determined m6A sites exhibit similar regulatory patterns, we stratified them into two classes: RRACH-only sites and ARE-flanking m6A sites (i.e., RRACH motifs with at least one ARE within ± 50 nucleotides). Differential expression analysis using DESeq2 showed a clear distinction between the two m6A subclasses

in activated T cells (Fig. 2E), with RRACH-flanking ARE sites associated with significantly shorter half-lives compared to RRACH-only sites (Supplementary Fig. S2F). These short-lived transcripts (GLORI-cluster i) showed a 29% overlap with miCLIP-cluster 2, characterised by strong miCLIP-ARE and miCLIP-RRACH signals (Fig. 2A), and included immunoregulatory mRNAs such as *CXCR4, CXCR6, G3BP2, IL7R, IL10RA, LSM1, TNFAIP3, TNFAIP8, and ZFP36L2* (Supplementary Data 1). Although the GLORI-cluster ii displayed similar half-life distributions when compared to the noGLORI group (Supplementary Fig. S2F), it still overlapped by 19% with miCLIP-cluster 2 and included several short-lived mRNAs such as *TNF* and *TNFRSF1B* (Supplementary Data 1). These findings suggest that ARE-flanking m6A sites and isolated m6A sites are functionally distinct in terms of mRNA stability during T cell activation. Moreover, the two m6A subclasses may contribute differently to post-transcriptional regulation relevant to T cell immune functions.

To test for associations between these m6A signatures and T cell subtype-specific roles, we interrogated mRNAs half-lives within transcriptional modules known to be co-activated in effector versus memory T cells[56]. Noticeably, the gene modules with shorter half-lives in the SLAM-seq data comprised late effector- and memory-related mRNAs (Supplementary Fig. S2G - modules C, D and F), as defined by Best et al.[56] and these were the most destabilised mRNAs during CD8+ T cell activation (Supplementary Fig. S2H). We next grouped mRNA half-lives based on their quartile distribution, < 2.3 (25%), 2.3−3.5 (50%), 3.5−5.3 (75%) and > 5.3 h (100%) in Day5 activated T cells and analysed how they connect to both m6A signatures (miCLIP and GLORI clusters) and CD8+ gene modules (Fig. 2G). Indeed, the late effector, memory precursor and naïve/late effector/memory genes (modules C, D and F) contained more shorter-lived (< 2.3 and 2.3−3.5 hr) than long-lived (3.5−5.3 and > 5.3 hr) mRNAs, which were enriched in miCLIP-cluster 2 and GLORI-cluster i (Fig. 2G – red and green lines). In contrast, the control groups (nomiCLIP and noGLORI) contained mainly cell cycle/division genes, which were overall longer lived (> 2.3 hr) (Fig. 2G – blue lines). These data suggest that ARE-flanking m6A sites in the 3′UTR is a feature of highly unstable – or meta-unstable – mRNAs with T cell memory roles in the activated state.

This led us to assess the most unstable (< 3.0 hr, 30 p^th) and stable (> 5.3 hr, 30 p^th) mRNAs in our SLAM-seq data based on the motif-specific m6A signatures. To this end, we trained conditional random forest (cRF) models to classify mRNAs as unstable ($N = 230$) or stable ($N = 230$) based on eight experimental features in our miCLIP dataset: (1–6) cluster identity, i.e., presence or absence of mRNA in each miCLIP cluster (c1 to c4) or GLORI cluster (ci and cii); (7) CD8+ state (noAct, Day1 and Day5) as a categorical variable; (8) CD8+ gene modules (A to G) as a categorical variable; and two 3′UTR features, (9) frequency of RRACH pentamers and (10) frequency of ARE (WWAWW) pentamers. First, we determined that two t-distributed stochastic neighbour

embedding (t.SNE) dimensions could partly resolve unstable and stable mRNAs based on the ten features considered (Fig. 2H), suggesting that a binary classifier may perform well in predicting the two mRNA classes. To determine the best predictive features of unstable and stable mRNAs, we used a conditional variable importance measure that computes the change in prediction accuracy (number of mRNAs classified correctly) when each predictor is permuted in a conditional manner (i.e., using a within-groups permutation that accounts for correlated variables)[57]. By determining the conditional variable importance on the 30 cRF models (3 repeats of 10-fold resampled iterations each) in the training set (75% of mRNAs), we showed that CD8[+] gene modules, miCLIP-cluster 2 and ARE frequency in the 3'UTR were among the most important features in its unique contribution to the cRF models (Fig. 2I). Noticeably, the frequency of ARE motifs within 3'UTRs varied more substantially across clusters than did the frequency of RRACH motifs (Supplementary Fig. S2I). Particularly, GLORI-cluster i was enriched for AREs, whereas GLORI-cluster ii was comparatively depleted of AREs relative to noGLORI mRNAs (Supplementary Fig. S2I).

To evaluate the performance of the cRF classifier, we assessed the predictions from the i) 10-fold cross-validation (CV) during training (training performance) and ii) final model on a held-out test set (25% of mRNAs) in terms of the following metrics: accuracy, specificity, precision, recall (or sensitivity), positive predictive value (PPV), negative predictive value (NPV) and F1 measure. The test set performance of the cRF classifier was 75–83% for each metric, indicating good predictive performance for the positive class (unstable mRNAs) (Fig. 2J – green points). Thus, the cRF classifier showed strong performance in predicting truly unstable mRNAs (PPV), and the high specificity indicates accurate identification of both unstable and stable mRNAs.

In addition, the balanced F1 score on the test set (78%) highlights the model's overall effectiveness in balancing sensitivity and precision in the classification of unstable mRNAs (Fig. 2J), with CD8[+] gene modules and interlinked AREs being important features in this prediction (Fig. 2I). Across the 10-CV folds, the training performance was 65–67% for each metric on averaged (Fig. 2J), which suggests that the 30 cRF models can be improved by being trained on more comprehensive data. Importantly, the receiving operating characteristic (ROC) and precision-recall (PR) curves confirmed that all 10-CV folds performed above a random classifier at all classification cutoff points (Fig. 2K, L), confirming the performance consistency across folds. Similarly, the ROC and PR curves from the final test predictions confirmed the generalisation capacity of the model (Fig. 2K, L). While the combined predictors did not fully account for mRNA instability in the test set (balanced accuracy = 79%) (Fig. 2J), these data indicate that ARE-flanking m6A sites (miCLIP-cluster 2) are better predictors of meta-unstable mRNAs than RRACH-only m6A sites (miCLIP-cluster 3 and GLORI-cluster ii). These findings support a model in which the presence of AREs within 3'UTR enhances m6A-dependent mRNA decay during T cell activation.

## Assessment of m6A-interlinked ARE sites on CD8[+] T cell essential mRNAs

To confirm our transcriptomic findings on essential genes for CD8[+] T cel biology, we assessed the stability of newly transcribed mRNAs harbouring ARE-flanking m6A sites and RRACH-only m6A sites (Fig. 2A - cluster 2). To this end, we treated CD8[+] T cells with DMSO or METTL3 inhibitor and, on Day 5 after activation, we blocked de novo transcription with actinomycin D and measured mRNA decay over time. METTL3 inhibition increased mRNA stability for *IL7R* (T cell memory marker[58]), *TNF* (essential effector molecule) mRNAs and a trend increase in *CD69* (T cell activation marker[59]) (Fig. 3A and Supplementary Fig. S3A). METTL3 inhibition/deletion also showed an overall increase in total mRNA levels of several mRNAs linked to memory formation (e.g., *IL7R*) and chemokines and cytokines important for

CD8[+] T cell function (*CCL4, TNF, IFNG*) (Fig. 3B and Supplementary Fig. S3B, S3C).

IL7Rα is an essential receptor for CD8[+] T cells, whose surface expression serves as a marker for long-lived memory CD8[+] T cells[60]. Particularly, the *IL7R* mRNA showed high-confidence miCLIP-RRACH and miCLIP-ARE sites (Supplementary Fig. S3D), also detected by GLORI as an ARE-flanking m6A within 50nt of a RRACH (GAACU) sequence (Fig. 2E – cluster i). In agreement with previous results (Figs. 1 and 2), we determined higher *IL7R* mRNA stability and total mRNA levels upon METTL3 inhibition (Fig. 3A, B), as well as an increased population of CD8[+] T cells expressing this surface marker and with higher intensity (Fig. 3C–F). These data indicate that m6A levels in ARE-flanking RRACH sites contribute to destabilisation of meta-unstable mRNAs, such as *IL7R* ($t_{1/2}$ = 2.3 hr), that characterise memory-precursor functions in activated CD8[+] T cells (Fig. 2G – module D, Supplementary Data 1).

Other important genes for CD8[+] T cell function include cytokines such as TNFα and IFNγ, essential for target clearance but also highly toxic, thus requiring a fined-tuned regulation of their expression. Memory CD8[+] T cells accumulate pre-formed mRNAs of *IFNG* and *TNF*, which can be rapidly translated upon activation without the need of de novo transcription[61]. Both *TNF* and *IFNG* mRNAs exhibited miCLIP-AREs, and the *TNF* had a clear miCLIP-RRACH site (Supplementary Fig. S3E, F), with METTL3 inhibition causing an increase in their steady-state levels (Fig. 3B). We used GFP reporter assays to further investigate the effect of m6A on TNFα and IFNγ expression, where GFP was fused with either the 3'UTR of *IFNG*, *TNF* or *GzmB* mRNA, and the median fluorescence intensity (MFI) measured (Supplementary Fig. S3E–H). CD8[+] T cells were transduced with the constructs, rested, and treated with increasing concentrations of the FTO inhibitor meclofenamic acid (MA) for 7 h, followed by overnight CD8[+] T cell activation[62]. The *GzmB* mRNA was used as a negative control, as it harbours a short 3'UTR, a feature previously associated with enhanced stability in T cells[63]. This was confirmed by SLAM-seq, which showed a $t_{1/2}$ of 12 h in no Act and 11 h in Day 5 activated CD8[+] T cells (Supplementary Data 1). In addition, the *GZMB* 3'UTR contained only one high-confidence miCLIP-RRACH site in activated T cells (Supplementary Fig. S3G). As expected, FTO inhibition had no effect on the proportion of GFP-positive cells (Supplementary Fig. S3I). Noticeably, the MFI was significantly reduced in cells expressing the 3'UTR-*IFNG* and 3'UTR-*TNF* constructs upon treatment with MA (Fig. 3G and Supplementary Fig. S3J). In contrast, GFP levels from the 3'UTR-*GzmB* construct remained unaltered by FTO inhibition (Fig. 3G and Supplementary Fig. S3J–K). In line with these results, secretion of endogenous IFNγ and TNFα proteins was also decreased following FTO inhibition during CD8[+] T cell activation (Fig. 3H–I), without compromising overall T cell viability (Supplementary Fig. S3L). Although the *GzmB* 3'UTR did contain a putative miCLIP-ARE site, its signal intensity was comparable to that of the input control site, suggesting this site is not reliably associated with m6A modification (Supplementary Fig. S3G). These results indicate that m6A methylation promotes the decay of short-lived cytokine mRNAs, thereby attenuating effector protein expression in activated CD8[+] T cells.

## Interplay of miCLIP-ARE sites in the control of *TNF* mRNA levels

The impact of m6A on RNP assembly may depend on the additional regulatory motifs beyond the RRACH, including AREs. To independently validate whether ARE-flanking m6A sites can be directy m6A modified, we applied the single-base elongation and ligation-based qPCR amplification (SELECT) method, which exploits the decreased activity of DNA polymerases and nick ligation to m6A bases[64]. We focused on the 3'UTR of the *TNF* mRNA, given the importance of the cytokine TNFα for CD8[+] T cell function[3,7] and because it contains a clear miCLIP-ARE site (AUUUAUUA) located within 15 nucleotides of a RRACH (AGACA) motif (Supplementary Fig. S3E). We designed SELECT

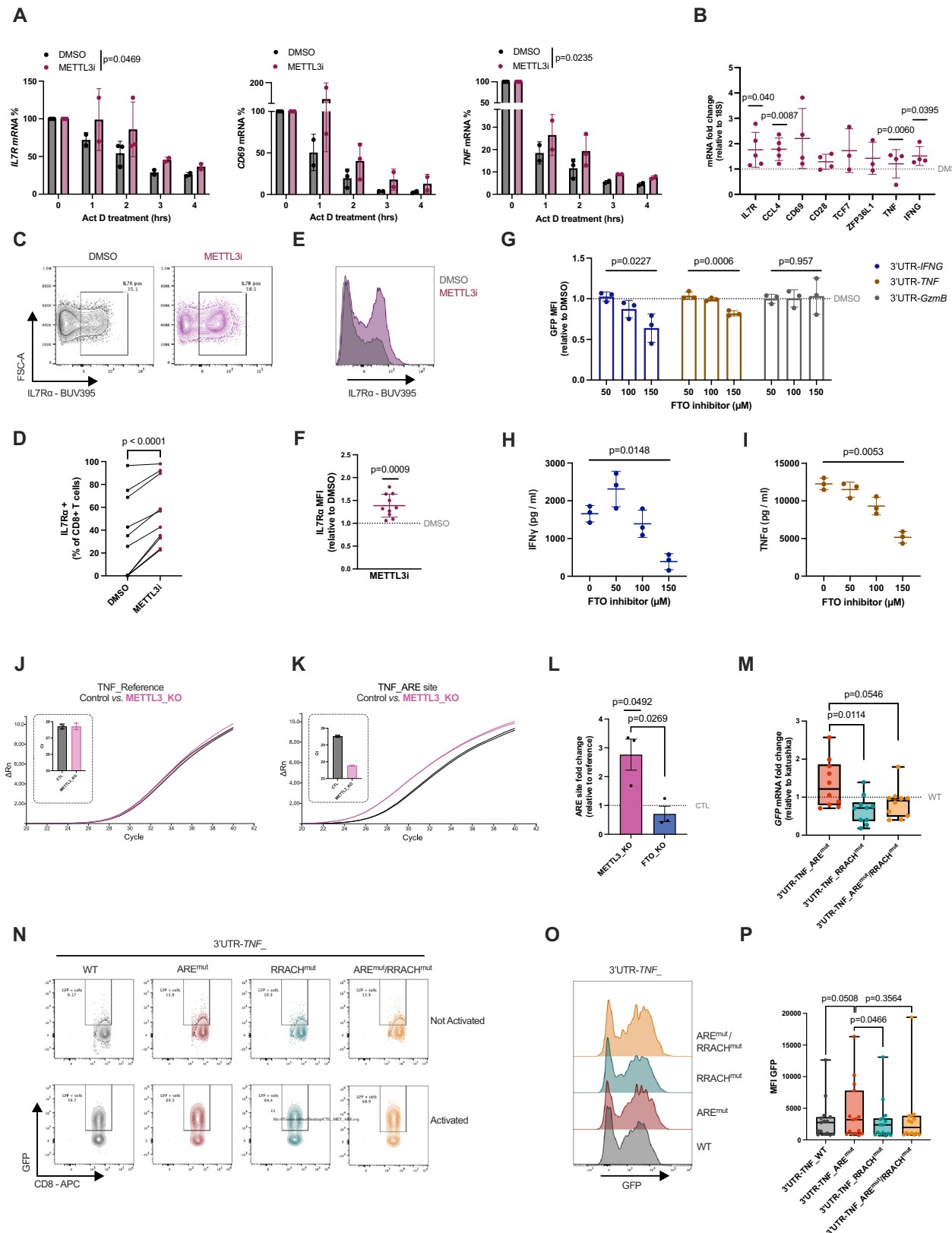

primers for the *TNF* 3'UTR that targeted the miCLIP-ARE site and a distal unmodified adenosine (A) as a reference site, located within 5 nucleotides of a high-confidence miCLIP-RRACH site and 85 nucleotides away from the miCLIP-ARE site (Supplementary Fig. S3E). For SELECT experiments, we adjusted the amount of input RNA based on the qPCR amplification. We then compared the SELECT qPCR

amplification curves of primary human CD8+ T control cells and CD8+ T cells with knockout (KO) of either METTL3 methyltransferase (METTL3-KO) or FTO demethylase (FTO-KO) (Supplementary Fig. S3B). While we did not observe differences in amplification curves between control and METTL3-KO CD8+ T cells at the reference site (Fig. 3J), we found greater amplification at the target miCLIP-ARE site in

**Fig. 3 | Effect of ARE-interlinked m⁶A sites on mRNA stability. A** Actinomycin D experiments showing % of remaining mRNA in CD8⁺ T cells treated with DMSO or METTL3inhibitor. Mean ± SD; $n = 3$ (one donor provided material for 0,2 h). Mixed-effects model (REML), p-values indicate treatment effects. Donors were processed in parallel. Representative results from three independent experiments (*IL7R*, *TNF*). **B** mRNA levels *IL7R* ($n = 5$), *CCL4* ($n = 5$), *CD69* ($n = 4$), *CD28* ($n = 4$), *TCF7* ($n = 3$), *ZFP36L1* ($n = 3$), *TNF* ($n = 4$), *IFNG* ($n = 4$) in CD8⁺ T cells treated with MET-TL3inhibitor. Fold change relative to DMSO (grey line). Reference gene 18S. Mean ± SD; paired two-tailed *t* tests on ΔCt values. For *TNF*, one Grubbs-identified outlier is displayed but excluded from statistics. **C, D** Representative flow cytometry plots illustrating IL7Rα gating (**C**) and quantification of IL7Rα⁺CD8⁺ T cells (**D**) ($n = 10$). Paired two-tailed *t* tests. **E, F** Representative histogram (**E**) and quantification (**F**) of IL7Rα median fluorescence intensity (MFI) in IL7Rα⁺ cells treated with METTL3inhibitor relative to DMSO (grey line) ($n = 10$). Mean ± SD; one-sample two-tailed *t* test. **G** GFP MFI in T cells transduced with GFP-3′UTR constructs (S3H) after treatment with FTO inhibitor followed by PMA/ionomycin activation relative to DMSO ($n = 3$). One-way ordinary ANOVA. **H, I** IFNγ (**H**) and TNFα (**I**) protein levels measured by ELISA in DMSO (0 μM) or FTOinhibitor treated cells ($n = 3$). RM one-way ANOVA. **J, K** Representative amplification curves (ΔRn) and Ct quantification for reference (**J**) and miCLIP-ARE (**K**) amplicons in control (grey) and METTL3-KO (pink) samples ($n = 3$). Mean ± SD. **L** Fold-change expression (ΔΔCt) for the miCLIP-ARE site in METTL3-KO and FTO-KO cells relative to control (grey line). Mean ± SEM; unpaired two-tailed *t* test comparing METTL3-KO and FTO-KO; paired *t* test for control vs METTL3-KO on ΔCt values; ($n = 3$). **M–P** mRNA levels (**M**) and GFP protein levels (**N–P**) in CD8⁺ T cells transduced with GFP-3′UTR constructs (S3O). mRNA levels normalised to 3′UTR-*TNF*_WT (grey line) ($n = 10$); one-way ANOVA with Holm–Šídák correction (**M**). Representative plots of resting or PMA/ionomycin-activated cells (**N**). GFP MFI quantification in activated cells (**O, P**) ($n = 15$); paired two-tailed *t* tests. Box-and-whisker plots (min–max, median, 25th–75th percentiles). All n values reflect distinct donors. Source data are provided as a Source Data file.

METTL3-KO cells (Fig. 3K, L), indicating lower m⁶A levels in the absence of METTL3. Conversely, comparison of WT and FTO-KO CD8⁺ T cells revealed reduced amplification at the miCLIP-ARE site (Supplementary Fig. S3M, N). We also observed a significant difference in amplification between METTL3-KO and FTO-KO cells at this site (Fig. 3L). These results show that the changes detected in the amplification curves were m⁶A dependent and sensitive to both METTL3 and FTO expression. These data provide orthogonal evidence for the existence of METTL3-dependent m⁶A modifications at specific RRACH-flanking ARE sites, suggesting these composite motifs function as a distinct m⁶A regulatory unit. We then sought to determine the role of motif-specific m⁶A signatures identified in our miCLIP on transcript expression. We used the GFP reporter system to further investigate the role of each motif-specific miCLIP site in the 3′UTR of *TNF*. Specifically, we used either the GFP construct fused with the WT 3′UTR of *TNF* (3′UTR-*TNF*_WT) or with mutated forms of miCLIP sites (Supplementary Fig. S3O). The mutated *TNF* 3′UTRs we used had either a point mutation (A > G) at the miCLIP-ARE site (3′UTR-*TNF*_AREᵐᵘᵗ), a point mutation (A > G) at the miCLIP-RRACH site (3′UTR-*TNF*_RRACHᵐᵘᵗ) or two-point mutations at both sites (3′UTR-*TNF*_AREᵐᵘᵗ/RRACHᵐᵘᵗ) (Supplementary Fig. S3O). We then retrovirally transduced CD8⁺ T cells with the GFP 3′UTR-*TNF* constructs. To account for differences in trans-duction efficiency, we simultaneously transduced the CD8⁺ T cells with a construct that expressed Katushka under no 3′UTR control (Supplementary Fig. S3O). After transduction, the cells were rested (kept in culture for an additional 7–10 days), and the RNA levels of GFP and Katushka were measured. We found higher *GFP* mRNA levels in the presence of mutant miCLIP-ARE compared to mutant miCLIP-RRACH (Fig. 3M and Supplementary Fig. S3P). When both sites were mutated, we observed lower ($P = 0.0546$) mRNA levels compared to the mutant miCLIP-ARE (Fig. 3M and Supplementary Fig. S3P). To test the effect of the mutations on protein expression, we reactivated the transduced CD8⁺ T cells with PMA/ionomycin for 16 h and measured GFP levels before and after activation (Fig. 3N–P). As expected, GFP was unde-tectable in resting CD8⁺ T cells (Fig. 3N), although we could quantify *GFP* mRNA levels in the resting state (Fig. 3M). In contrast, following 16 h of activation, GFP protein levels increased across all conditions tested (Fig. 3N), with a higher intensity of GFP under mutant miCLIP-ARE conditions when compared to WT ($P = 0.0508$) and mutant miCLIP-RRACH (Fig. 3O, P). Katushka levels remained unchanged across conditions both before and after activation of transduced CD8⁺ T cells (Supplementary Fig. S3Q–T). These results showed that the RRACH-flanking ARE site enhances mRNA decay only in the presence of intact RRACH sites capable of m⁶A modification. Interestingly, mutation of the RRACH site alone was insufficient to alter *TNF* mRNA levels, contrary to expectations based on the established role of RRACH in mRNA destabilisation[9]. Collectively, these data lead to the hypothesis that the functional impact of m⁶A modifications may in some cases, depend on combinatorial recognition of methylated RRACH alongside its flanking ARE sites.

## miCLIP-AREs recruit YTHDF and other unique RNA processing factors

To investigate whether miCLIP-ARE sites modulate the recruitment of RBPs, we first examined the local enrichment of m⁶A in RNAs bound by specific ARE-binding proteins. We performed CLIP experiments in Jurkat T cells using antibodies against canonical m⁶A-readers (YTHDF1 and YTHDF2) and ARE-binding proteins relevant to T cell biology (HuR, ZFP36). Crosslinked RNA fragments were isolated and analysed by tandem mass spectrometry (LC–MS/MS) to quantify m⁶A and adeno-sine (A) levels (LC–MS/MS) (Supplementary Fig. S4A). Use of Jurkat T cells in these experiments provided sufficient material to repro-ducibly detect m⁶A ribonucleosides by LC-MS/MS, which could not be achieved with primary T cells due to lower RNA yield (see note in Methods). We found that the m⁶A/A ratio was elevated in RNAs bound to YTHDF1/2, HuR and ZFP36, compared to those bound by the polyA-binding protein (PAPB), which preferentially interacts with A-rich sequences. This indicates that m⁶A is enriched in RNA regions bound by both m⁶A-readers and ARE-binding RBPs. Of note, the input RNA (total crosslinked RNA) showed an intermediate m⁶A/A ratio relative to the PABP-bound RNAs (Supplementary Fig. S4A).

To assess how the presence of m⁶A influences RNA–protein interactions close to AREs, we performed quantitative mass spectrometry-based RNA interactome capture using synthetic RNA probes designed to mimic key sequence contexts identified by miCLIP and GLORI profiling, as previously described[65]. Specifically, we tested RNA oligonucleotides containing either the canonical m⁶A-modified RRACH motif (GGACU) or a composite motif in which a UUAUU (ARE) pentamer was immediately flanked upstream by a RRACH sequence. Unmethylated versions of each probe served as controls. The methy-lation was placed directly on the central adenosine of the ARE (AGACU-Um⁶AUU), inspired by the original miCLIP signal. Although GLORI data showed that m⁶A sites are located within RRACH rather than ARE, our probes nevertheless capture the composite architecture of ARE-flanking RRACH sites to allow comparison with RRACH-only probes. We performed a forward pulldown, wherein immobilised A and m⁶A probes were first incubated with human T cell lysates. After incubation and washes, bound proteins were digested and then differentially labelled with light (A bait pull-down) and medium (m⁶A bait pull-down) dimethyl isotopes, respectively. We also performed a label swap experiment, which serves as a replicate measurement (reverse). We determined the M/L ratio of bound proteins for the forward and reverse pulldowns, such that m⁶A-readers are depicted in the upper right quadrant (high forward M/L ratio; low M/L ratio in the reverse experiment) (Fig. 4A, B and Supplementary Fig. S4B). As a control, we confirmed that YTH domain-containing proteins (YTHDF1, YTHDF2,

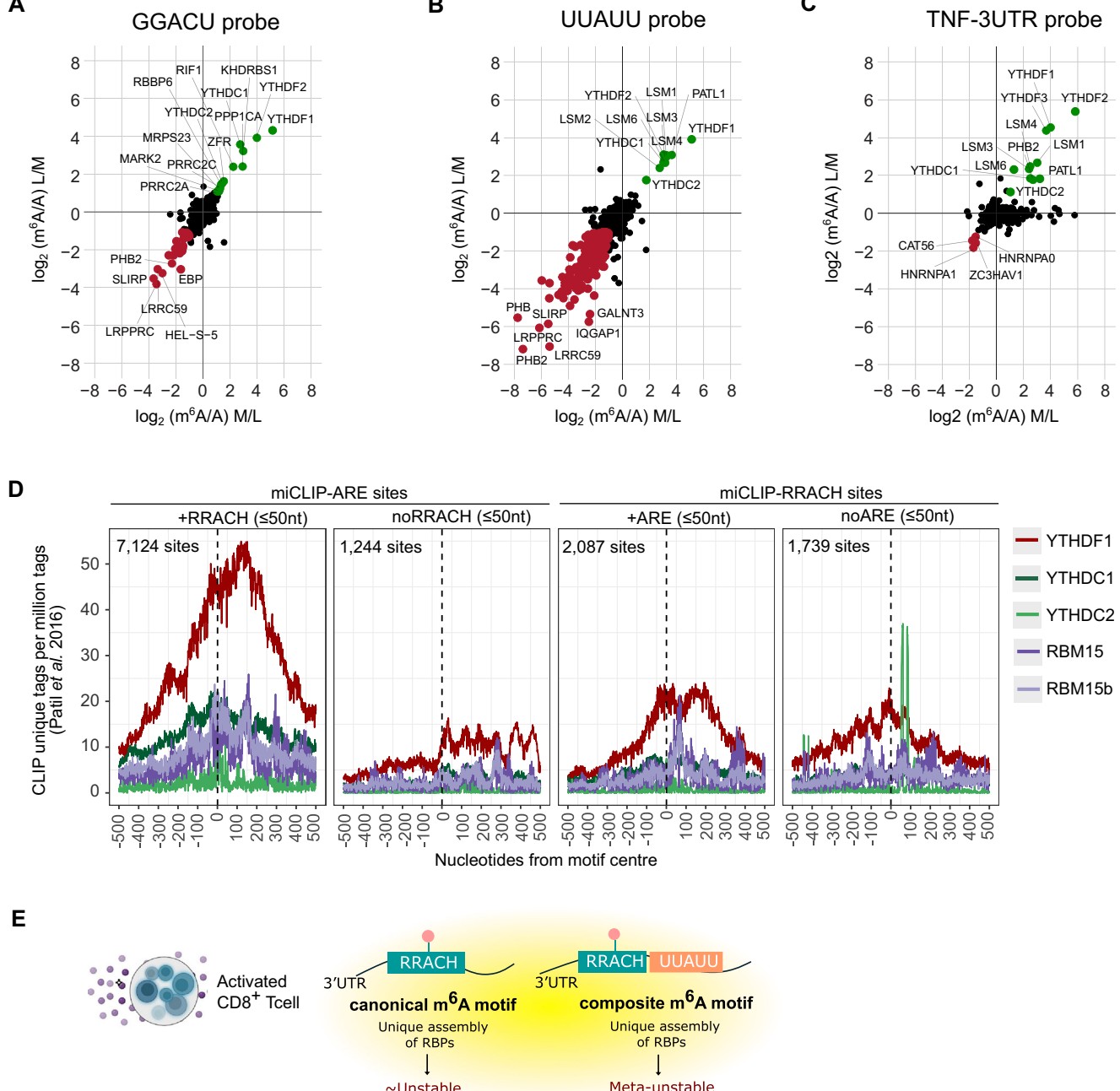

**Fig. 4 | Modulation of RNA-binding activity by miCLIP-ARE sites. A–C** Proteomic interactome screening using RNA pulldowns with m⁶A-modified and unmodified probes. Shown are log₂ fold-changes in protein abundance for RNA-binding proteins (RBPs) binding to: (**A**) an m⁶A-modified GGACU probe, (**B**) an m⁶A-modified UUAUU (ARE) probe, and (**C**) an m⁶A-modified region of the TNF 3′UTR. Fold-changes were calculated relative to the corresponding unmethylated probe. Data represent one representative experiment from pooled human T cells (*n* = 9 donors). **D** Metagene distribution of published CLIP data for m⁶A-reader proteins[67] at endogenous miCLIP-defined sites. Left: CLIP signal centred at miCLIP-ARE sites, stratified by whether a RRACH motif occurs within ± 50 nt. Right: CLIP signal

centred at miCLIP-RRACH sites, stratified by whether an ARE occurs within ± 50 nt. **E** Model illustrating how canonical (RRACH-only) versus composite (ARE-flanking m⁶A) motifs differentially regulate mRNA decay. Composite motifs, defined by m⁶A at RRACH elements positioned within ± 4 nt of ARE pentamers, promote rapid mRNA destabilisation during T cell activation. These composite sites recruit a distinct set of RBPs compared to canonical RRACH-only m⁶A sites, enabling selective decay of immunoregulatory mRNAs in activated T cells. Parts of panel (**E**) were created in BioRender. Foskolou, I. (2026) https://BioRender.com/ m18fugc. Source data are provided as a Source Data file.

YTHDC1 and YTHDC2) were more strongly bound to the m⁶A-modified GGACU probe than to the unmethylated GGACU probe (Fig. 4A), showing that one repeat of the m⁶A consensus sequence GGACU is sufficient to capture canonical m⁶A readers in T cells[65]. Conversely, we found that various members of the LSM family of RNA-binding proteins (LSM1, LSM2, LSM3, LSM4, and LSM6) specifically bound to m⁶A-modified UUAUU relative to the unmodified UUAUU probe

(Fig. 4B and Supplementary Fig. S4B). Interestingly, all the YTHDF paralogues were also significant readers of m⁶A-modified UUAUU sequences, whereas several more proteins were repelled by the m⁶A-modified UUAUU than by the unmodified GGACU probe (Fig. 4B and Supplementary Fig. S4B). It has been reported that LSM proteins assemble into RNPs that facilitate pre-mRNA splicing and mRNA degradation[66]. To test whether miCLIP-ARE sites recruit specific RBPs

in a biological sequence context, we performed pulldown experiments targeting the *TNF* 3'UTR, where the m[6]A-modified (and unmodified) probes contained a UUAUU pentamer surrounded by a sequence (15 nt upstream and downstream) corresponding to the respective region of the *TNF* 3'UTR. The *TNF* 3'UTR probe contained the miCLIP-ARE site flanked downstream by a RRACH site (UUm[6]AUUACCCCCUCCUUC AGACA, see complete probe in Methods), as assayed above (Supplementary Fig. S3E).

Similarly to Fig. 4B, YTH paralogues and the LSM family of proteins were found to bind specifically to the m[6]A-modified *TNF* probe (Fig. 4C and Supplementary Fig. S4C). Finally, to determine whether the high-confidence ARE-flanking m[6]A sites identified by miCLIP and GLORI profiling could serve as binding sites for canonical m[6]A readers, we checked the iCLIP data for YTHDF, YTHDC paralogues, RBM15 and RBM15b[67]. We observed that YTHDF1 binding sites, as defined by published unique iCLIP tags[67], were enriched at both miCLIP-ARE and miCLIP-RRACH sites, but not at control ARE or RRACH sites lacking miCLIP signal (Fig. 4D and Supplementary Fig. S4D, E). Control sites were expression-matched to the miCLIP sites by selecting 3'UTRs from the DESeq2 model that were expressed in CD8[+] T cells but showed low miCLIP signal for the respective motifs (Supplementary Fig. S4D). Surprisingly, unique YTHDF tags were also detected at miCLIP-CRACH sites, however, due to their low number, it is difficult to interpret this distribution (Supplementary Fig. S4D). Notably, YTHDF1 binding showed a prominent central peak at miCLIP-AREs flanked by RRACH (≤ 50nt) but not at isolated miCLIP-AREs (Fig. 4D). Enriched central peak was observed at both classes of miCLIP-RRACH sites, but we noticed somewhat broader crosslinking profile at RRACHs that were proximal to ARE motifs (Fig. 4D and Supplementary Fig. S4E). Collectively, these data show that m[6]A-modified AREs can recruit LSM proteins and YTHDF proteins, which may cooperatively alter mRNA decay through YTHDF-dependent pathways. These findings support a model in which RRACH-flanking AREs form composite m[6]A motifs that assemble distinct RNP complexes to more strongly promote mRNA destabilisation in activated CD8[+] T cells as compared to the RRACH-only m[6]A sites (Fig. 4E).

## Discussion

The degradation of mRNAs is an essential aspect of the regulation of transcript numbers[68]. This is ever more important for dosage-sensitive genes, such as cytokines, for which small fluctuations in rates of mRNA decay can have significant effects on mRNA and protein abundance[6]. In T cells, the tight regulation of cytokine expression is required to ensure early and transient responses, since aberrant cytokine production can lead to severe autoimmune diseases, while insufficient cytokine levels cause impaired antitumor responses[3]. Here, through a motif discovery analysis and supervised learning approach on miCLIP/GLORI and SLAM-seq data, we have identified ARE-flanking m[6]A motifs that mark immunoregulatory mRNAs for rapid decay in activated CD8[+] T cells.

In this study, we apply an improved miCLIP protocol, motif discovery analysis (PEKA) and GLORI profiling to find that m[6]A-modified RRACHs are often flanked by AREs (Fig. 1C, D). The closest arrangement is a RRACH sequence (e.g., GAACU) directly next to an ARE (e.g., AUUUA) to form a hypothetical AUUUAGAACU or GAACUAUUUA motif. The sequence required for METTL3-dependent m[6]A methylation[44,48] can thus partly overlap with an ARE-like sequence, likely an atypical class III ARE[4].

The molecular effects of m[6]A-modified RRACHs on mRNA degradation range from subtle to profound[9,53]. Understanding the specificity of this process would enable more precise identification of m[6]A variants that regulate basic cell functions. We reasoned that a low frequency of m[6]A sites in non-RRACH motifs does not indicate a lack of biological importance, since cellular responses may require subsets of mRNAs whose stability depends on different regulatory mechanisms. CD8[+] T cell activation is a crucial process that relies on the tight control

of cytokine and inflammatory mRNA levels and is largely achieved by ARE-binding proteins[3,7]. However, how ARE-mediated mRNA decay is integrated with other mechanisms, including RNA modifications, to define T cell functions is not understood. Through a supervised learning approach on miCLIP/GLORI and SLAM-seq data, we revealed that CD8[+] T cell activation leads to faster decay of mRNAs harbouring ARE-proximal m[6]A modifications. We refer to these highly unstable transcripts as meta-unstable mRNAs, reflecting their dynamic reduction in half-life. This reduction occurs predominantly upon CD8[+] T cell activation. This group includes key essential mRNAs for CD8[+] T cell function (Fig. 2A – cluster 2, 2E – cluster i, Supplementary Data 1). Notably, the meta-unstable mRNAs were more accurately predicted by ARE-proximal m[6]A sites than by remaining m[6]A sites. This composite m[6]A motif, defined by RRACH-flanking ARE sites, may constitute a distinct regulatory unit for m[6]A-dependent mRNA decay. We tested this model using orthogonal approaches and targeted assays focused on key immunoregulatory genes. We employed SELECT studies to add orthogonal evidence of the presence of m[6]A in a specific RRACH-flanking ARE site in the *TNF* 3'UTR (Supplementary Fig. S3E). GFP reporter assays showed that a single-nucleotide mutation of the ARE site of 3'UTR-*TNF* led to increased mRNA levels only in the presence of an intact RRACH site, but not when mutating both the RRACH and ARE sites. It should be noted that the reporter assays are insufficient to fully disentangle m[6]A-dependent effects from other sequence-dependent influences on mRNA levels (Fig. 3). Notably, some ARE-binding RBPs, such as HuR[69] may be able to bind to both the wild-type ARE (UUAUU) and its mutant (UUGUU) sequences in the reporter. On the other hand, it is possible that the A > G mutation in the 3'UTR-*TNF*_ARE[mut] may impair the binding of ARE-binding RBPs regardless of m[6]A status. Nevertheless, our findings show that mRNA meta-instability of the *TNF* and other immunoregulatory mRNAs likely depends on the interplay between RBPs binding to m[6]A sites and their flanking ARE motifs.

We present two lines of evidence that ARE-flanking m[6]A sites and canonical RRACHs have different efficacy on mRNA decay CD8[+] T cells, likely due to the formation of somewhat distinct m[6]A-dependent ribonucleoprotein (RNP) complexes. First, using cellular extracts of T cells, we show that ARE-flanking m[6]A sites recruit YTHDF paralog proteins along with additional RBPs, especially the LSM proteins[66]. Importantly, LSM4, a highly enriched RBP according to our interaction screening, was recently shown to bind *TNF* mRNA in T cells after activation[70]. Second, analysis of YTHDF1/2/3 binding profiles[67] showed that YTHDF paralogs show a somewhat broader crosslink profile to the composite sites with AREs flanked by the methylated RRACH sites when compared to isolated RRACH sites, consistent with *Arabidopsis* iCLIP data showing that m[6]A-modified RRACH motifs are favoured in pyrimidine-rich regions[71]. While all three YTHDF paralogs are known to contribute to the degradation of m[6]A-modified mRNAs, two studies have indicated that YTHDF2 plays a main role in mRNA decay through two pathways: (1) the endoribonucleolytic-cleavage pathway via RNaseP/MRP[72] and (2) the CCR4/NOT deadenylation pathway[73]. However, whether and to what degree the RnaseP/MRP and CCR4/NOT pathways operate differently on meta-unstable mRNAs with different m[6]A signatures has not been addressed here. Moreover, whether m[6]A-interlinked ARE and RRACH sites work cooperatively or in opposing ways on specific mRNAs was not determined, and merits further consideration.

Beyond the molecular basis of mRNA decay, our study links m[6]A to meta-unstable mRNAs that are involved in memory and late effector functions (Fig. 2)[56,74]. Since m[6]A and TNFα play important roles in antitumor immunity, it will be relevant to assess whether ARE sites functionally linked to m[6]A influence T cell differentiation and activity. Taken together, the findings support a model whereby ARE-flanking m[6]A sites recruit distinct regulatory RNP assemblies that enable selective decay of memory precursor and late effector transcripts in T cells. This context-dependent m[6]A code reveals how combinatorial

RNA motifs can fine-tune transcript stability in response to activation cues in CD8[+] T cells. Further understanding of the mechanisms mediating the combinatorial recognition of m[6]A and its flanking ARE sites could lead to valuable approaches for fine-tuning the immune system.

## Methods

### Reagents and resources

All reagents (chemicals, antibodies, bacterial strains, cell lines, biological samples, peptides, recombinant DNA, oligonucleotides and CRISPR-Cas9 crRNAs) are provided in the "Key Resources Table" in Supplementary Information.

### Human T cells

Human CD8[+] T cells were isolated either directly after blood donation (8-12 h after blood collection) or cryopreserved and used after cryopreservation. PBMCs were isolated through Ficoll-Paque PLUS density gradient separation (GE Healthcare). The cells were incubated in 21% oxygen and 5% carbon dioxide at 37 °C. Sex, gender or age were not determined for any of the donors.

For miCLIP and SLAM-Seq experiments, human CD8[+] T cells were isolated with MACS Miltenyi kits (total CD8[+] T cells: 130-096-495; total CD8[+] MicroBeads: 130-045-201; and total FcR-blocking: 130-059-901) following the manufacturer's instructions. CD8[+] T cells were activated with aCD3/CD28 beads (1:1 bead-to-cell ratio) (11132D, Gibco) in fresh complete RPMI media (52400-025; Gibco) containing 10% FBS, 1% penicillin–streptomycin and IL-2 (30 µ/ml; Roche).

For GLORI experiments, human CD8[+] T cells were isolated with the MACS Miltenyi kit (total CD8[+] MicroBeads: 130-045-201) following the manufacturer's instructions. CD8[+] T cells were either kept non-activated for 1 day or were activated with aCD3/CD28 beads (1:1 bead-to-cell ratio) (11132D, Gibco) in fresh complete RPMI media (52400-025; Gibco) containing 10% FBS, 1% penicillin–streptomycin and IL-2 (30 µ/ml; Clinigen) for 1 or 5 days. Non-activated and Day1 activated cells were treated for 24 h with DMSO or 4 µM METTL3 inhibitor (MedChemExpress, cat. no. HY-134836), while Day5 activated T cells were treated with the same inhibitor for 5 days. For Day 5 activated samples, beads were removed at day 4 and fresh medium was replenished twice during culture.

For the CRISPR/Cas9 and SELECT experiments, human CD8[+] T cells were isolated with MACS Miltenyi kits (total CD8[+] T cells: 130-096-495; total CD8[+] MicroBeads: 130-045-201; and total FcR-blocking: 130-059-901) following the manufacturer's instructions. CD8[+] T cells were activated in 24-well plates precoated overnight at 4 °C with 2 µg/ml anti-mouse IgG2a (BioLegend) in phosphate-buffered saline (PBS). The plates were washed and coated for > 3 h with 1 µg/ml a-CD3 (HIT3a, Biolegend) at 37 °C. The CD8[+] T cells were then seeded in plates supplemented with 1 µg/ml soluble a-CD28 (CD28.2; Biolegend) in fresh complete RPMI media (52400-025; Gibco) supplemented with 10% FBS, 1% penicillin–streptomycin and IL-2 (30 µ/ml; Clinigen).

For RNA pull-down and GFP construct transduction experiments, human T cells (both CD4[+] and CD8[+]) were activated in 24-well plates that were precoated overnight at 4 °C with 2 µg/ml anti-mouse IgG2a (BioLegend) in phosphate-buffered saline (PBS). The plates were washed and coated for > 3 h with 1 µg/ml a-CD3 (HIT3a, Biolegend) at 37 °C. The CD8[+] T cells were then seeded in plates supplemented with 1 µg/ml soluble a-CD28 (CD28.2; Biolegend) in fresh complete RPMI media (52400-025; Gibco) supplemented with 10% FBS, 1% penicillin–streptomycin and IL-2 (30 µ/ml; Clinigen). For GFP construct transduction experiments, the number of GFP-positive or Katushka-positive CD8[+] T cells was determined via flow cytometry. For the RNA pull-down experiment, T cells were isolated from 9 donors, activated and expanded separately for 7 days. On day 7, the cells were pooled in three falcons (3 donors in each pool). After centrifugation, the cells were resuspended in RIPA buffer (150 mM NaCl, 50 mM Tris (pH 8), 1 mM EDTA, 10% glycerol, 1% NP-40, 1 mM DTT, and 1 x protease inhibitor),

incubated on a rotator at 4 °C for 2 h, and centrifuged, after which the supernatant was collected in new tubes that were snap frozen.

For inhibitor treatments, T cells were treated with either DMSO or 6 µM METTL3 inhibitor (4 µM for GLORI datasets) (MedChemExpress, cat no. HY-134836) for 4 to 5 days. Media refreshment was done every 1-2 days. For the FTO inhibitor, T cells were treated with DMSO or increasing concentrations of Meclofenamate[62] (sodium salt, Cayman Chemical, Item No. 70550) for 6-7 h before 16 h activation with PMA/ ionomycin (for GFP reporter assays) or treated overnight (18 h) before 4 h activation (for ELISA assays).

### Cloning

For the Katuska construct, the pMIG-w (Addgene) plasmid was used as a backbone, and the Katushka sequence was integrated by using the NcoI and PacI restriction enzymes. For GFP 3′UTR-IFNγ, 3′UTR-TNFα and 3′UTR-GzmB plasmids, the full-length 3′UTR of each gene was amplified from human genomic DNA and cloned into BamHI and NotI sites of pRETRO-SUPER GFP downstream of GFP, as previously described[70]. The single and double mutants were generated by site-directed mutagenesis via Twist Bioscience (https://www.twistbioscience.com). The sequences were verified by sequencing.

### Retroviral transduction of human T cells

The viral supernatant was collected after transfecting FLYRD18 cells (retroviral packaging cells) with the GFP-3′UTR or Katushka plasmids using GeneJammer as a transfection reagent. The viral supernatant was collected 2 days after transfection. T cells were isolated and activated as described above. After 2 days of activation, T cells were co-transduced with retrovirus containing one of the GFP-3′UTR constructs or with retrovirus containing the Katushka construct at a ratio of 1:1. Nontissue culture–treated 24-well plates were coated overnight with 50 µg/ml RetroNectin (Takara) and washed once with PBS prior to the addition of both viral supernatants. The plates were centrifuged for 30 min at 4 °C and 2820 x g. The viral supernatant was removed, and 0.5 × 10[6] T cells were added to each well. The cells were harvested 1-2 days after transduction and cultured at a concentration of 1 × 10[6] cells/ ml for an additional 6–8 days. Rested cells (not reactivated) were collected for RNA isolation and RT–PCR analysis. For flow cytometry analysis of protein levels, cells were either reactivated with PMA (10 ng/ ml) or ionomycin (1 µM) for 16 h or left non-activated. Flow cytometry was performed as described below.

### Quantitative real-time PCR (qRT–PCR)

Total RNA was extracted from isolated T cells using TRIzol (Invitrogen), Direct-zol RNA Miniprep Kits (Zymo Research, R2052), or Quick-RNA Miniprep Kit (Zymo Research, R1055), and 300–500 ng of RNA was used for cDNA synthesis (SuperScript III, Invitrogen or Maxima First Strand cDNA Synthesis Kit for RT-qPCR, K1641, Thermo Fisher). All kits were used according to the manufacturer's instructions. Samples were run in technical triplicates. qRT–PCR was performed using SYBR Green on a StepOne Plus system (Applied Biosystems). Ct values were normalised to HPRT or 18S levels. The primer sequences can be found in the "Key Resources Table".

### Actinomycin D treatment

Peripheral blood mononuclear cells (PBMCs) were thawed, and CD8[+] T cells were isolated using a CD8[+] isolation kit (CD8[+] MicroBeads, 130-045-201, Miltenyi biotec). Isolated cells were activated using anti-CD3/CD28 activation beads (Dynabeads™ Human T-Activator CD3/ CD28 for T Cell Expansion and Activation, 11131D, ThermoFisher Scientific) and cultured in T cell medium (RPMI 1640 supplemented with L-glutamine and 25 mM HEPES (ThermoFisher Scientific, cat no. 52400-025) supplemented with 10% FCS, 10,000 µ/ml penicillin-10 mg/ml streptomycin (Sigma, cat no P4333), and human IL-2 (30 µ/mL). Cells were treated with either DMSO or 6 µM METTL3

inhibitor (MedChemExpress, cat no. HY-134836) for 4 or 5 days. Media refreshment was done every 1-2 days. Activation beads were removed on day 4. On day 5, cells were treated with 5 µg/mL actinomycin D (Sigma-Aldrich, Cat. No A9415) for 0, 1, 2, 3, or 4 hrs to assess RNA stability. Subsequently, cells were washed with cold PBS, centrifuged, and the resulting pellets were snap-frozen and stored at − 80 °C until further analysis. RNA was extracted as mentioned above and qRT-PCR followed for quantification of mRNA levels.

### ELISA
Secreted IFNγ and TNFa levels of CD8$^+$ T cells were determined using ELISA. Assays were performed with uncoated ELISA kits (Catalog No. 88-7316-88; 88-7346-88 ThermoFisher Scientific), according to the manufacturer's instructions. Absorbance was determined using a microplate reader (Sunrise, Tecan) at a wavelength of 450 nm.

### Genetic modification of T cells with Cas9 RNPs
crRNAs were designed in Benchling (https://benchling.com) and are listed in the "Key Resources Table". The Alt-R crRNA and tracr-RNA were reconstituted to 100 µM in nuclease-free duplex buffer (Integrated DNA Technologies). As a negative control, the nontargeting negative control crRNA #1 was used (Integrated DNA Technologies). Oligos were mixed at equimolar ratios (i.e., 4.5 µl of total crRNA + 4.5 µl of tracrRNA) in nuclease-free PCR tubes and denatured by heating at 95 °C for 5 min in a thermocycler. The nucleic acids were cooled to room temperature prior to mixing with 30 µg of TrueCut Cas9 V2 (Invitrogen) to produce Cas9 ribonuclear proteins (RNPs). The mixture was incubated at room temperature for at least 10 min prior to nucleofection. For nucleofection, human CD8$^+$ T cells were activated for 48 h as described above. Cells were electroporated in 16-well strips in a 4D Nucleofector X unit (Lonza) with the programme EH100 for human T cells with P2 Primary Cell buffer (Lonza). Knockout efficiency was determined on days 7–10 after electroporation by Western blot.

### SELECT
The elongation and ligation-based qPCR amplification method SELECT was used as previously described[64]. Specifically, CD8$^+$ T cells were isolated from healthy individuals, activated and cultured as described above. On day 2 after activation, the cells were genetically modified with CRISPR/Cas9 as described above and further cultured for an additional 8–12 days. The amount of RNA for each sample and each target was previously verified by quantitative real-time PCR (qRT–PCR), and the same amount of RNA was used for SELECT (normalised to the WT control). Specifically, to validate a presumed m$^6$A site, two primers (up and down primers) were designed to flank the site of interest via Benchling (https://benchling.com). A nonm$^6$A site on the same mRNA was also tested as a reference control (Reference Up and Down primers). The sequences of all primers used are listed in the "Key Resources Table". Total RNA (1.5 µg) was used per experiment. The RNA was mixed with 40 nM Up Primer, 40 nM Down Primer and 5 µM dTTP in a total of 17 µl of 1 × CutSmart buffer (NEB). The RNA and primers were annealed by incubating the mixture at a temperature gradient of 90 °C for 1 min, 80 °C for 1 min, 70 °C for 1 min, 60 °C for 1 min, 50 °C for 1 min, and 40 °C for 6 min. Subsequently, 3 µl of a mixture containing 0.01 U of Bst 2.0 DNA polymerase, 0.5 U of SplintR ligase and 10 nmol of ATP was added to the former mixture to a final volume of 20 µl. The final reaction was incubated at 40 °C for 20 min, denatured at 80 °C for 20 min and kept at 4 °C. Quantitative real-time PCR (qPCR) was subsequently performed with the primers qPCR_F and qPCR_R for SELECT (200 nM; sequences at "Key Resources Table") and with PowerUp SYBR Green on a StepOne Plus system (Applied Biosystems). For the qPCR, 2–4 µl of the elongation and ligation samples were used. qPCR was performed under the following conditions: 95 °C, 5 min; 95 °C, 10 s; 60 °C, 35 s, 40 cycles; 95 °C, 15 s; 60 °C, 1 min; 95 °C, 15 s (for which the fluorescence was collected at a ramping rate of

0.05 °C/s); and 4 °C. The data were analysed by Design & Analysis 2 (DA2) Software v2.6.0 (Thermo Scientific).

### Flow cytometry
The cells were pelleted by centrifugation and stained with antibodies in FACS Buffer (5% FBS, 2 mM EDTA in PBS) at 4 °C for 30–60 min. The stained cells were washed with FACS buffer 2–3 times and processed for flow cytometry. The emission spectra "spillover" were corrected by compensation using compensation beads (01-1111-41; OneComp eBeads) mixed with each fluorescent probe. The BD LSR-Fortessa and BD FACSymphony flow cytometers and the Sony Spectra Analyser were used. The flow cytometry data were analysed using FlowJo (BD Biosciences, version 10). The antibodies used can be found in the "Key Resources Table".

### Western blotting
Human CD8$^+$ T cells were isolated, activated and engineered by CRISPR/Cas9 as described above. A total of $3 \times 10^6$ cells were collected, washed and pelleted. The cell pellets were incubated with urea-tris buffer (8 mol/L urea, 50 mmol/L Tris-HCl (pH 7.5), 150 mmol/L β-mercaptoethanol) and centrifuged at $14,000 \times g$ and 4 °C for 15 min. The proteins were separated via SDS–PAGE and subsequently transferred to PVDF membranes. The membranes were then blocked in 5% milk prepared in PBS supplemented with 0.05% Tween 20 and incubated with primary antibodies overnight at 4 °C and HRP-conjugated secondary antibodies for 1 h at room temperature the next day. Following ECL exposure (Invitrogen), the membranes were imaged using an iBrightCL1000 (Thermo Fisher). The antibodies used can be found in the "Key Resources Table".

### miCLIP library preparation
We applied miCLIP (methylation individual-nucleotide-resolution crosslinking and immunoprecipitation)[25,26] to profile m$^6$A using the latest improved iCLIP protocol (iiCLIP)[36] to produce cDNA libraries with increased efficiency and minimal loss of material. Compared to previous (m)iCLIP versions, this protocol features (i) the use of DMSO (and no DTT) for improved ligation efficiency; (ii) infrared imaging of the RNA:anti-m$^6$A antibody complex, as in irCLIP[75]; (iii) reverse transcription (RT) with SSIV and RT primers that contain carbon spacers, as in irCLIP; (iv) ampure bead-based purification of cDNAs; and (v) circularisation with commercial circligase II with betaine additive. The miCLIP protocol is completed from UV crosslinking to amplify cDNA libraries within 4 days.

RNA extraction and fragmentation (Day 0): Total RNA was extracted from CD8$^+$ T cells using the mirVana isolation kit according to the manufacturer's instructions (Thermo Fisher Scientific, #AM1560), purified polyadenylated RNA was extracted using two rounds of poly(A) tail hybridisation with Oligo-dT magnetic Dynabeads (Thermo Fisher Scientific, #61002), and fragmented (750–1000 ng) in a buffered zinc acetate solution (pH 5.3) at 70 °C for 15 min (pH 5.3).

Immunoprecipitation (IP), UV crosslinking and SeqRv adaptor ligation (Day 1): The fragmented polyadenylated RNA was subjected to IP using an anti-m$^6$A antibody (Ab) (Abcam, catalogue no. ab151230) conjugated to protein A/G Dynabeads (Thermo Fisher Scientific, catalogue no. 88803) for 2 h at 4 °C. The RNA:Ab complex was irradiated twice with 150 mJ/cm2 UV at 254 nm, and the cDNA library was prepared as described previously[36]. After IP, the (protein A/G) beads were washed twice in high-salt buffer (50 mM Tris-HCl, pH 7.4, 1 M NaCl, 1 mM EDTA, 1% Igepal, 0.1% SDS, 0.5% sodium deoxycholate) and twice in PNK (20 mM Tris-HCl, pH 7.4, 10 mM MgCl$_2$, 0.2% Tween-20) buffer. The samples were incubated at 37 °C for 40 min for 3' end dephosphorylation; the beads were washed with 1 x ligation buffer (50 mM Tris-HCl (pH 7.5) and 10 mM MgCl$_2$), and 3' end ligation to an infrared dye-conjugated and biotinylated L3 adaptor (SeqRv) was performed on the beads overnight at 16 °C with shaking at 150 g.

SeqRv adaptor removal, SDS–PAGE and RNA purification (Day 2): The 3' end-ligated RNA:Ab complex on the beads was washed twice in high-salt buffer and twice in PNK wash buffer, and the excess SeqRv adaptor was removed by resuspending the beads in a mixture of 5' deadenylase (NEB) and RecJ$_f$ exonuclease (NEB) for 30 min at 30 °C followed by 30 min at 37 °C while shaking at 150 g. The beads were washed twice with high-salt buffer and once with PNK buffer, and the protein–RNA complexes were eluted in 1x NuPAGE loading buffer (supplemented with 100 mM DTT) at 70 °C for 1 min. The protein–RNA complexes were separated via SDS–PAGE in a 4–12% Bis–Tris gel (Invitrogen), transferred to a nitrocellulose membrane for 2 h at 30 V, and visualised via infrared imaging. The RNA:Ab complex was excised from the membrane, treated with proteinase K (which leaves only a short peptide at the crosslinking site) for 1 h at 50 °C with shaking at 150 g, and the RNA was purified by adding a (25:24:1) mixture of phenol:chloroform:isoamyl alcohol (Sigma–Aldrich). The mixture was subsequently transferred to a prespun 2 ml Phase Lock Gel tube and incubated for 5 min at 30 °C with shaking at 150 g. The samples were centrifuged for 5 min at 21,000 × g, chloroform was added to the top (aqueous) phase of the biphasic solution, and the tubes were inverted to mix and centrifuged for 5 min at 21,000 × g. The aqueous layer was transferred into a new tube, and the RNA was precipitated overnight in 70% ethanol containing 300 mM sodium acetate and 1 μl of glycogen blue (15 mg/ml).

Reverse transcription, cDNA purification and circularisation (Day 3): The RNA pellets were washed twice in 70% ethanol, resuspended in water and reverse transcribed into cDNA with barcoded RT primers (which contained a sequence complementary to the SeqRv adaptor) using the Superscript IV (SSIV) Kit. To minimise technical variations in library preparation, we multiplexed up to four CLIP libraries after the RT step using barcoded RT primers. We did not perform RT primer removal because this step was not tested or incorporated into the iiCLIP protocol at the time of this study. The cDNAs were purified using Agencourt AMPure XP beads, circularised using CircLigase II with betaine additive, purified again using the AMPure XP beads, eluted in water and stored at − 20 °C.

PCR amplification (Day 4) – The cDNA was amplified using an optimal number of cycles in PhusionHF Master Mix containing P3/P5 Illumina sequencing primers.

We applied a similar protocol to generate input libraries, in which 750–1000 ng of polyadenylated RNA was fragmented, directly processed for 3' end dephosphorylation, converted into cDNA and amplified as described above.

## miCLIP data analysis

The miCLIP and input data were uploaded and analysed on the FlowBio platform (https://app.flow.bio) using a standardised iCLIP analysis described therein[76]: demultiplexing (https://app.flow.bio/pipelines/447171969201517470/) followed by the CLIP-Seq (https://app.flow.bio/pipelines/960154035051242353/) pipeline. Using the CLIP-Seq pipeline as described previously https://docs.flow.bio/docs/clip-pipeline-1.0/, (1) the 3' end of the reads were trimmed for the 3' end adaptor sequence (AGATCGGAAGAGCGGTTCAG) by TrimGalore; (2) the reads were pre-mapped to an index of rRNA and tRNA sequences using Bowtie; (3) the unmapped reads were mapped to the human genome (GRCh38) using STAR; (4) the aligned reads were deduplicated based on the unique molecular identifiers (UMI) and start position; (5) the start position minus one in uniquely mapped reads was used as a proxy for the crosslink position, i.e., the most likely site of the methylated base[25,47], and crosslink bed files were produced; and (6) the peaks were determined using Paraclu and Clippy peak callers. The GRCh38 GENCODE annotation was used for all human data. Other summary files reporting total crosslink counts in genes and RNA biotypes and quality control metrics presented as a MultiQC report were generated by the CLIP-Seq pipeline; all files from the miCLP and Input

samples are retrievable at https://app.flow.bio/projects/663588932565340992/. We applied positionally enriched k-mer analysis (PEKA) to extract motifs that were enriched in proximity to high-confidence crosslink sites, i.e., 'thresholded crosslinks', located within crosslink peaks and with high counts relative to low-count crosslink sites located outside the peaks[42]. We used crosslinks and peaks (generated by Paraclu or Clippy) as inputs for PEKA; we ran PEKA by setting the proximal window to 10 nt (-w 10), the distal window to 150 nt (-dw 150), the k-mer length to 5 nt (-k 5), the smoothing window to 6 nt (-s 6), the percentile parameter for assignment of thresholded crosslinks to 0.7 (-p 0.7), and without subsampling of foreground and background crosslinks (https://github.com/ulelab/peka). Each k-mer at the "thresholded crosslink" position was assigned a PEKA score[42], and the top 20 k-mers were plotted based on the PEKA scores and P-value of less than 0.01 (Fig. 1B, Supplementary Fig. S1B). We defined "PEKA crosslinks" as the number of crosslinks that overlap with the genomic positions of 'thresholded PEKA crosslinks'. To check the alignment at motif centres (Fig. 1), the PEKA crosslinks were normalised at each meta position in the 3'UTR by the total number of crosslinks in the 3'UTR. We used PEKA crosslinks in the 3'UTR for all other analyses (Fig. 2) unless otherwise noted.

## GLORI library preparation and data analysis

Human CD8[+] T cell pellets from 6 donors were lysed in TRIzol, and total RNA was isolated using Direct-zol RNA Miniprep Kits (Zymo Research, cat. no. R2052) according to the manufacturer's instructions. RNA quality was assessed with a Bioanalyzer (Agilent). The RNA of donors was pooled in 2 (not activated) or 3 (Day 1 and Day 5 activated) samples. GLORI libraries were prepared using the NEBNext Ultra II Directional RNA Library Prep Kit for Illumina (New England Biolabs, cat. no. E7760S). The protocol uses a dUTP-based method for second-strand synthesis, resulting in strand-specific libraries in which Read 2 corresponds to the sense strand of the original RNA. Library preparation was carried out according to the manufacturer's instructions, and sequenced on an Illumina NovaSeq 6000 by the Advanced Sequencing Facility at The Francis Crick Institute. Downstream analyses were performed using the second read (L5_2) of each pair, which contains the chemically informative read for m⁶A detection. Adaptor trimming was performed using Trim Galore (v0.6.7) with the following parameters: -q 20 --stringency 1 -e 0.3 --length 20. Reads shorter than 20 nucleotides after trimming were discarded. The remaining high-quality reads were aligned to an *A > G converted* version of the human genome (GRCh38) using GLORI-tools (https://github.com/liucongcas/GLORI-tools), which accommodates the chemical reactivity-based profiling of m⁶A through misincorporation signals. Strand-specific pileups for all four nucleotides (A, C, G, T) were generated directly from the resulting BAM files. Per-base coverage and A-to-G or T-to-C conversion ratios were computed and exported as both BigWig and BedGraph files to facilitate visualisation in IGV or related genome browsers. Conversion ratios were calculated as A/(A + G) for the forward strand and T/(T + C) for the reverse strand. To identify putative m⁶A sites, pileup data were processed in R using the GenomicRanges package, focusing on positions with reference A or T located within ± 2 kb of annotated protein-coding genes (Ensembl release 113). These positions were further annotated for overlap with canonical m⁶A motifs (GAC, GAAC) and their reverse complements. Transcript-level mapping was performed using the annotatePeakInBatch() function from ChIPpeakAnno, enabling transcript-level annotation. Given that a single site can map to multiple transcripts or genes, redundancy was handled by tracking unique "position_id" fields to identify and filter duplicate entries. Additional genomic context was assigned using genomicElementDistribution() (ChIPpeakAnno), classifying sites at three hierarchical levels: transcript-level regions (promoter, downstream, gene body), exon–intron boundaries, and exon subtypes (5'UTR, CDS, 3'UTR, or other noncoding exons within protein-coding genes). To increase

specificity, we applied two filtering criteria: (i) a minimum read depth of ≥15 for A + G (or T + C on the reverse strand) and (ii) a minimum base conversion ratio (A/(A + G) or T/(T + C)) of ≥ 0.1. The resulting datasets were saved as GRanges objects in.rds format for downstream meta-analysis in R. Supporting files, including gene annotations in GTF format, motif coordinates, genomic distribution plots, and conversion rate boxplots, were made for visualisation and further analysis. This integrated pipeline enabled the identification and annotation of GLORI-defined m⁶A sites across the CD8⁺ T cell transcriptome.

## SLAM-Seq library preparation and determination of mRNA half-lives

SLAM-Seq experiments were conducted as previously described[52] with certain modifications. Specifically, CD8⁺ T cells were isolated from two healthy donors as described above. The cells were either not activated or activated with aCD3/CD28 beads (1:1 bead-to-cell ratio). Both non-activated and activated cells were cultured in fresh complete RPMI media supplemented with 10% FBS, 1% penicillin–streptomycin and IL-2 (30 μ/ml). Approximately $10 \times 10^6$ cells were used per condition. For non-activated and day 1 activated cells, 100 μM 4SU (T4509; Sigma) was added to the culture medium every 4 h for 24 h in total. The cells were then collected and washed with warm media. After centrifugation, the cells were plated in fresh RPMI media supplemented with 10% FBS, 1% penicillin–streptomycin, IL-2 and 100 mM uridine (U3750; Sigma). After the addition of uridine, the cells were further cultured for 0, 0.5, 1, 3, 6 and 16 h. At each of the above time points, $1\text{-}2 \times 10^6$ cells were harvested, the activation beads were removed, and the RNA was isolated. For day 5 of activation, CD8⁺ T cells were activated for 4 days, after which the activation beads were removed, and 4SU was added as described above. After 24 h of 4SU, the cells were washed, further cultured with uridine and harvested at 0, 1, 3, 6 and 16 h. Total RNA was extracted from human CD8⁺ T cells after each chase time point using TRIzol reagent, followed by column-based purification. RNA integrity was assessed by Bioanalyzer (Agilent). SLAM-seq libraries were generated as previously described[52], with minor modifications. In brief, thiol-modified RNA was chemically alkylated to introduce T > C mutations detectable by NGS. Libraries were prepared from total RNA using the Lexogen QuantSeq 3′ mRNA-Seq Library Prep Kit FWD, V2 (Cat. No. 191.96, Lexogen GmbH), which produces forward-stranded Illumina-compatible libraries enriched for the 3′ ends of poly-adenylated transcripts. The workflow consists of oligo(dT)-primed reverse transcription from the poly(A) tail, selective degradation of RNA, and second-strand synthesis primed by random oligonucleotides carrying Illumina linker sequences to preserve strand specificity. Final cDNA libraries were PCR-amplified, purified, quantified, assessed for quality using an Agilent Bioanalyzer, and sequenced on an Illumina HiSeq 4000 by the Advanced Sequencing Facility at The Francis Crick Institute. To calculate mRNA T > C conversions, the analysis pipeline was implemented using Snakemake 5.3.0[77]. Sequencing reads were demultiplexed with Cutadapt 1.18, with no allowed mismatches in the barcodes, quality-filtered with a cutoff of Q20, trimmed of the barcodes (cut 12), and short reads were discarded (length 16). Poly-adenylated stretches (> 4) at the 3′ end were trimmed to 20 tandems without indels or mismatches. The trimmed reads were aligned to the reference human genome (GRCh38) using the Slamdunk map 0.3.3[78] with local alignment, allowing up to 100 multimapping reads (-n 100). Aligned reads were filtered to retain only those with a minimum identity of 95% and at least 50% of the read bases mapped. Reads ambiguously mapping to more than one 3′ UTR were discarded using the UTR annotation pipeline (https://github.com/AmeresLab/UTRannotation). SNPs were identified using VarScan 2.4.1[79] with default parameters, establishing a 10-fold coverage cutoff and a variant fraction cutoff of 0.8 (-c 10-f 0.8). T > C conversions overlapping with SNPs required a minimum base quality (Phred score) of 26. T > C conversions were counted in rolling windows, normalised by T content

and coverage of each position, and averaged per possible transcript 3′ UTR. The quality of the T > C conversions was assessed using Slamdunk Alyoop 0.3.3[78]. To calculate mRNA half-lives, T > C conversions were background-subtracted, normalised to the chase uridine treatment and fitted to an exponential decay model using the R package minpack.lm 1.2[80]. The mRNA half-lives that correlated with the model ($R^2 > 0.5$) under all the experimental conditions were determined by $t_{1/2} = \ln(2)/k$, where $k$ is the decay constant in the mono-exponential model. Using this approach, we determined accurate half-lives for 4158, 5585 and 5866 transcripts in activated CD8⁺ T cells on day 1 and day 5, respectively.

## Motif-specific analysis of miCLIP, GLORI and input counts and half-life classification

The total number of PEKA crosslinks per 3′UTR was normalised using the DESeq2 model (DESeq factor), transformed using the variance stabilisation function (vst) and standardised (centred and scaled). The top 1000 3′UTRs with the highest variability of DESeq2-normalised counts across samples were clustered based on the Euclidean distance of counts and plotted on the heatmap. DESeq2 normalisation was applied to the number of PEKA crosslinks in each motif class (RRACH or WWWWW pentamers; a 'W" centre was considered for DESeq2 analyses, instead of 'A'), summed across the 3′UTR and to the corresponding input counts, i.e., input reads that started at the same coordinates, also summed across the 3′UTR. We excluded four input replicates from the DESeq2 model (noAct, replicate 1 and 2; Day1, replicate 1; and Day5, replicate R3), due to the low number of counts (< 500) in the 3′UTR. All the miCLIP replicates (13 WWAWW and 13 RRACH samples) were considered for DESeq2 analysis. The same procedure was carried out for differential analysis of GLORI-determined m⁶A counts using the DESeq2 model. To classify mRNA half-lives, we considered all 3′UTRs in DESeq2 clusters that had reliable half-lives and selected the most unstable (< 3.0 h, 30 p^th) and most stable (> 5.3 h, 70 p^th) mRNAs. Using the caret package (6.0.94) in R, we trained a conditional random forest (cRF) model to classify unstable ($N = 230$) and stable ($N = 230$) mRNAs based on the following miCLIP features: 1–6) cluster identity, where the presence (1) or absence (0) of mRNA in each DESeq2 cluster was coded as a dummy variable (using the ifelse() function), 7) the CD8⁺ state (noAct, Day1 and Day5) and 8) CD8⁺ gene modules (A–G) as categorical variables, and two genomic features: 9) frequency of RRACH pentamers and 10) frequency of ARE (WWAWW) motifs. We generated 30 conditional random forest (cRF) models by randomly splitting the training set (75% of mRNAs) into 10 subsamples ($k$ folds) of the same size (10-fold cross-validation) and computing the performance metric (accuracy) for each fold (k), with the remaining (k-1) of the folds used as training data; this process was repeated 3 times. The control parameters for the caret training were defined using the 'caret::trainControl' function, with the main arguments (method = "repeatedcv", number = 10, search = "random", repeats = 3, $p = 0.75$, sampling = "down", summaryFunction = twoClassSummary, classProbs = TRUE), and the cRF models (train object) generated using the 'caret::train' function with arguments (method = 'cforest', trControl = fit.control, metric = "Accuracy", maximise = T, tuneLength = 10), with 'fit.control' representing the R object with the control parameters defined above, and the number of predictors (mtry) was tuned for optimal model accuracy. The variable importance in the cRF models was determined using the 'caret::varImp' function with arguments (scale = T, conditional = T). The cross-validation performance of the cRF models was determined using the 'caret::confusionMatrix' function. The final class predictions of the cRF models were made by applying the "stats::predict" function to the test set (25% of the mRNAs). We generated cRF models to classify desta-bilised (log2-fold < − 0.73) and stabilised (log2-fold > 0.30) mRNAs in Day5-activated CD8⁺ T cells using the same approach, and the cross-validation performance was determined as above.

## Quantification of m⁶A levels in RNP complexes (CLIP-MS)

For CLIP–MS analysis of RBP-bound ribonucleosides, Jurkat T cells were UV-crosslinked, processed by CLIP, and the RNA was isolated and hydrolysed to ribonucleosides for LC-MS/MS analysis, as described below. The detected m⁶A ion counts were in the 0.5–5 fmol range, with signal-to-noise ratios of 15–30 relative to blanks. Although RBP abundance and regulation differ quantitatively between Jurkat and primary T cells, their qualitative RNA-binding profiles are broadly conserved across cell types (given similar expression)[81], supporting the use of Jurkat T cells as a reductionist model for mechanistic studies of RBP binding and m⁶A methylation. The RNA:RBP complexes for the 4 RBPs (HuR, ZFP36L1, YTHDF1 and PABP) were purified from Jurkat T cells using the iiCLIP protocol, with the following specifications. Jurkat T cells were cultured in twelve 15 cm culture plates (3 plates per RBP tested) until 80% confluence, irradiated twice with 0.15 J/cm2 UV at 254 nm, harvested by scraping, and centrifuged at $500 \times g$ at 4 °C for 5 min. The cell pellets were subsequently lysed in iCLIP lysis buffer (50 mM Tris-HCl, pH 7.4, 100 mM NaCl, 1% Igepal, 0.1% SDS, 0.5% sodium deoxycholate) supplemented with complete protease inhibitor cocktail (Roche). To generate RNA fragments of optimal size and digest genomic DNA, we incubated 9 mg (at 1 mg/ml) of Jurkat T cell lysates with 1 µ/ml of RNase 1 and 2 µl of Turbo Turbo DNase (Invitrogen) in CLIP lysis buffer for 3 min at 37 °C with shaking at 150 g. The cell lysates were centrifuged for 10 min at $21,000 \times g$, and the supernatants were immunoprecipitated with 5 µg of antibody conjugated to protein A/G Dynabeads for 3 h at 4 °C. We kept 150 µl of supernatant as an 'input' control; this mixture was not subjected to IP and was directly purified by SDS–PAGE. The RNA:RBP complexes were purified by SDS–PAGE and excised from the membrane as described above (miCLIP-Day2), and the isolated RNA was digested into nucleosides, after which the m⁶A levels were quantified by LC–MS/MS[82]. In brief, we digested the purified RNA into ribonucleosides using one unit of nuclease P1 in 25 µl of 25 mM NaCl, 2.5 mM ZnCl₂ and 10 mM NaCH₃COO (pH 5.3) and incubated for 2 h at 37 °C. Subsequently, NH₄HCO₃ (100 mM) and 5 units of alkaline phosphatase were added, and the sample was incubated for 2 h at 37 °C. Formic acid was added at 0.1% v/v in a final volume of 50 µl, the samples were filtered (0.22 µm, Millipore), and 15–20 µl was analysed in duplicate by LC–MS/MS. The ribonucleosides were separated using a C18 reversed-phase column (100 × 2.1 mm, 3 µm particle size; Chromex Scientific) and detected by a TSQ Quantiva Triple Quadrupole mass spectrometer (TSQ Quantiva, Thermo Scientific) operating in positive ionisation mode with a capillary voltage of 3500 V, a sheath gas flow rate of 7.35 l/min, and a gas temperature of 325 °C. The retention time, mass transitions (*m/z*) and collision energies were ~4.1 min, 268.1 - > 136.1 *m/z* and 20 V, respectively, for adenosine and ~ 8.7 min, 282.1 - > 150.1 *m/z*, and 20 V, respectively, for m⁶A. A mixture of ribonucleoside standards of each ribonucleoside was run for absolute quantification. The data were recorded using Xcalibur 4.0.27 software (Thermo Fisher Scientific), and the areas under the adenosine and m⁶A peaks were determined using Skyline.

## RNA pulldowns with dimethyl labelling

RRACH, ARE, and TNFα pulldowns were performed with dimethyl labelling[65,83]. All buffer solutions were prepared with RNase-free reagents. For each RNA pulldown experiment, four individual pulldowns were performed. For each individual pulldown, 20 µl of streptavidin Sepharose high-performance slurry beads (Cytiva, 50% v/v) was used. The beads were washed with RNA-binding buffer (RBB, 50 mM HEPES-HCl (pH 7.5), 150 mM NaCl, 0.5% NP40 (v/v), and 10 mM MgCl₂) and centrifuged. All pulldown centrifugation steps were performed at 1,500 × g for 2 min at 4 °C. The beads were then incubated with the RNase inhibitor RNasin plus (Promega) in RNA binding buffer (100 µl of RBB with 0.8 units of RNasin/µl) for 15 min on ice. After centrifugation, the beads were pre-blocked with yeast tRNA (Invitrogen, 100 µg tRNA

in 1 ml RBB) for 1 h at 4 °C on a rotation wheel, followed by one wash with RBB. Then, the beads were incubated with either a control or m⁶A probe at a final concentration of 0.67 µM in 600 µl of RBB for 30 min at 4 °C while rotating. The probe sequences used for the RNA pulldowns can be found in the "Key Resources Table" (oligonucleotides). After probe incubation, the beads were washed once with RBB and twice with protein incubation buffer (PIB, 10 mM Tris-HCl (pH 7.5), 150 mM KCl, 1.5 mM MgCl₂, 0.1% (v/v) NP-40, 0.5 mM DTT, and complete protease inhibitors (Roche)). The beads containing the immobilised RNA probes were then incubated with 1 mg of whole-cell lysate pooled from 9 donors in 600 µl of PIB supplemented with 0.4 units of RNasin/µl and 30 µg of tRNA for 30 min at RT, followed by 90 min at 4 °C. Unbound and unspecific proteins were removed by washing three times with PIB followed by two 1 x PBS washes. The remaining liquid was removed with a syringe. For on-bead digestion, 50 µl of elution buffer (100 mM Tris, pH 8.5; 2 M urea; and 10 mM DTT) was added to the beads, which were subsequently incubated for 20 min at RT in a thermoshaker at 200 g. To prevent the reformation of disulphide bonds, iodoacetamide was added to a final concentration of 50 mM, and the mixture was incubated for 10 min in the dark at RT ($200 \times g$). This was followed by partial digestion of proteins from the beads by the addition of 250 ng of trypsin and incubation for 2 h at RT at 200 g. in the dark. After centrifugation, the supernatant containing the cleaved peptides was collected, and the remaining peptides were collected by adding 50 µl of new elution buffer to the beads and incubating them for 5 min at 200 g before the supernatant was added to the previously collected supernatant. Next, 100 ng of trypsin was added to the eluates, and the proteins were further digested overnight at RT. The eluted peptide samples were then prepared and desalted for mass spectrometry analysis via the C18 StageTip method[84]. StageTips were prepared by placing a double C18 plug in a 200 µl pipette tip free of polymers. The disk was then washed with 50 µl of methanol, 50 µl of buffer B (0.1% formic acid, 80% acetonitrile), and two washes of 50 µl of buffer A (0.1% formic acid). All washes were performed by centrifugation at $1500 \times g$, after which the StageTips were placed in adaptors suitable for centrifugation and collection of the flowthrough. The eluted peptides were then loaded on the StageTips and centrifuged at $1000 \times g$ for 10 min to allow the peptides to bind to the disk and the liquid to flow through. Peptides were then labelled by either light or medium (0.2% CH2O or 0.2% CD2O in 10 mM NaH2PO4, 35 mM Na2HPO4 and 32 mM NaBH3CN) dimethyl isotopes by labelling with 150 µl of the isotope labelling buffer and centrifugation of $1000 \times g$ for 5 min twice[83].

## Mass spectrometry (MS) analysis of RNA pulldown samples

After peptide elution from the StageTips, individual peptide samples were combined, resulting in forward and reverse reactions, which served as two technical replicates: *the forward reaction* included the control probe CH₂O (light) combined with the m⁶A probe CD₂O (medium); *the reverse reaction* included the control probe CD₂O (medium) combined with the m⁶A probe CH₂O (light). Elution of both the forward and the reverse reactions was performed using 30 µl of buffer B (0.1% formic acid, 80% acetonitrile), followed by an additional elution using 10 µl. After speedvac centrifugation to reduce the sample to 5 µl, 7 µl of buffer A (0.1% formic acid) was added. For the measurements, 5 µl of each sample was loaded on an Easy-nLC1000 connected online to an Orbitrap Exploris 480 (Thermo Fisher Scientific), and the data were acquired in a 60 min gradient. A 43 min acetonitrile gradient (12% to 30%) was used, followed by washes at 60% and 95% for a total of 60 min of data acquisition. The spray voltage was set to 2100 V in positive ion mode. The scans were collected in a data-dependent mode and collected in the top 20 data-dependent acquisition modes. The dynamic exclusion was enabled and set to 45 s. The Orbitrap resolution for a full scan was set to 120,000 in a scan range of 350–1300 m/z.

## Mass spectrometry analysis of protein abundance

The RNA pulldown results were processed using MaxQuant 1.6.6.0[85]. The DimethylLys0 and DimethylNter0 labels were selected, as were DimethylLys4 and DimethylNter4 (light and medium, respectively). Peptide identifications were restricted to tryptic peptides with no more than two missed cleavages. The minimal peptide length was set at 7. Carbamidomethyl cysteine was set as a fixed modification, and oxidised methionine and protein N-acetylation were searched as variable modifications. Database searching was performed with a main-search precursor mass tolerance of 4.5 ppm and a fragment ion mass tolerance of 20 ppm. FDR was controlled at 1% at the peptide-spectrum match (PSM) and protein levels. The re-quantification function was enabled, and the UniProt human FASTA database from June 2017 was used as a search database. Perseus v1.6.15.0[86] was used to filter and analyze the data. Proteins identified as identified only by site, reverse and potential contaminants were removed. Finally, proteins were filtered if at least two peptides were identified with at least one unique peptide. To identify and visualise significant interactors, forward and reverse ratios were log2 and -log2 transformed, respectively, and a ratio of both forward and reverse of 1 was used as a cutoff. Scatter plots were generated in R.

## Quantification and statistical analysis

The results are shown as the mean ± SEM or mean ± SD as stated in the figure legends. Statistical analysis was performed using R or Prism-10 software (GraphPad). $p < 0.05$ indicated statistical significance, and the statistical tests applied are stated in the figure legends.

## Ethical approvals

Peripheral blood mononuclear cells (PBMCs) were obtained from healthy donors at Cambridge Bioscience, National Health Service (NHS) Blood and Transplant (NHSBT: Addenbrooke's Hospital, Cambridge, UK) or Sanquin (Amsterdam, NL). The study was performed according to the Declaration of Helsinki (seventh revision, 2013). Ethical approval was obtained from the Eastern England-Cambridge Central Research Ethics Committee (06/Q0108/281), and consent was obtained from all the subjects. Written informed consent was obtained (Cambridge Bioscience, Cambridge, UK; NHSBT Cambridge, UK; Sanquin Research, Amsterdam, NL).

## Reporting summary

Further information on research design is available in the Nature Portfolio Reporting Summary linked to this article.

## Data availability

The data supporting the findings of this study are available from the corresponding authors upon request. The data generated in this study are publicly available. Specifically:

- miCLIP - sequencing data ArrayExpress (E-MTAB-15643); sequence raw and processed files: https://app.flow.bio/projects/663588932565340992/
- GLORI - sequencing data ArrayExpress (E-MTAB-15649); sequence raw and processed files: https://app.flow.bio/projects/127418738102891967/
- SLAM-Seq - sequencing data ArrayExpress (E-MTAB-15648); sequence raw and processed files: https://app.flow.bio/projects/783560763419435859/ (SLAM-Seq).
- The mass spectrometry proteomics data have been deposited to the ProteomeXchange Consortium via the PRIDE[87] partner repository with the dataset identifier PXD059083.
- Flow cytometry, western blots and analysed data are available at Mendeley Data: Mendeley Data, V1, https://doi.org/10.17632/bhgn4bn5ks.1

- Source data for the figures and Supplementary Figs. are provided as a Source Data file. Source data are provided in this paper.

## Code availability

Custom R scripts used to process crosslink data and generate metaprofiles shown in Fig. 1, 4D and Supplementary Fig. S1, S4D were developed for this study. The scripts execute standard data loading, normalisation, and plotting functions using publicly available R packages. Given their simplicity and reliance on existing libraries, the code is not deposited in a public repository but is available from the corresponding author upon request.

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

## Acknowledgements

We thank Richard Mitter for support with GLORI data analysis, Karen Davey, Natasja Kragten, Aurélie Guislain, Benoît P Nicolet and Ken GC Smith for valuable discussions and technical assistance; the Bioinformatics and Biostatistics Team and the Advanced Sequencing Facility (ASF) at the Francis Crick Institute for expert support; the Cambridge NIHR BRC Cell Phenotyping Hub; the flow cytometry facility from the School of the Biological Sciences; and the flow cytometry facility at Sanquin. This work was supported by the Wellcome Trust (103760/Z/14/Z) and the European Research Council (ERC; FP7/2007-2013/ERC grant agreement ID: 617837) awarded to J.U.; Wellcome Trust Principal Research Fellowship (214283/Z/19/Z), a Knut and Alice Wallenberg Scholar Award, the Swedish Medical Research Council (Vetenskapsrådet 2019-01485), the Swedish Cancer Fund (Cancerfonden, CAN2018/808), and the Swedish Children's Cancer Fund (Barncancerfonden PR2020-007) awarded to R.S.J.; the HORIZON-MSCA grant (701730) awarded to P.A.G.; and the HORIZON-MSCA grant (101107176) awarded to I.P.F. Work in the lab of M.V. is supported by an NWO-VICI grant (VI.C.202.013). The laboratories of M.W. and M.V. are part of the Oncode Institute, which is partly funded by the Dutch Cancer Society.

## Author contributions

Y.A.B. and A.A.M.M. contributed equally to this work. Conceptualisation, P.A.G., I.P.F., R.S.J., and J.U.; Methodology, P.A.G., I.P.F., Y.A.B., A.A.M.M., M.C.W., R.S.J., and J.U.; Software, P.A.G., K.K., I.R.dlM., N.D.Z., and R.F.; Investigation, P.A.G., I.P.F., Y.A.B., A.A.M.M., K.K., I.R.dlM., N.D.Z., Z.H., V.K., and R.F.; Writing-Original Draft, P.A.G. and I.P.F.; Writing-Review & Editing, all authors; Funding Acquisition, P.A.G., I.P.F., R.S.J., and J.U.; Supervision, I.P.F., M.V., M.C.W., R.S.J., and J.U.; Project Administration, P.A.G., I.P.F., M.C.W., R.S.J., and J.U.

## Funding

## Competing interests

The authors declare no competing interests.
