## [Transparent Peer Review file · Nature Communications]

Meta-unstable mRNAs in activated CD8⁺ T cells are defined by interlinked AU-rich elements and m6A mRNA methylation

Corresponding Author: Professor Jernej Ule

Version 0:

Reviewer comments:

Reviewer #1

(Remarks to the Author)

Reviewer Comments:

This manuscript explores the correlation between N6-methyladenosine (m6A) and mRNA decay in human CD8 T cells using the miCLIP method. The authors report that the combined miCLIP signal for RRACH and ARE motifs is enriched in the 3'-UTR of meta-unstable mRNAs during CD8 T cell activation. They also investigate the roles of these motifs in TNF mRNA stability and claim that YTHDF1 binds to methylated AREs of meta-unstable mRNAs.

Strengths:

1. The authors constructed a valuable dataset using sophisticated techniques, offering a new classification of mRNAs in human CD8 T cells based on m6A modification sites and instability.
2. A variety of data analysis methods were applied, providing important insights into post-transcriptional gene regulation, particularly in the context of m6A and mRNA decay.

Weaknesses:

1. The study does not adequately explore the physiological impact of these motifs on CD8 T cell function. The functional relevance of the observed m6A modifications remains unclear.
2. The authors focus on TNF mRNA, which may not be the most critical for CD8 T cell function. A broader investigation into other mRNAs that are more central to CD8 T cell biology would strengthen the study's conclusions.
3. Consequently, while the findings are of interest within the field of RNA-binding proteins and post-transcriptional regulation, the immunological relevance is limited. This may reduce the manuscript's impact from an immunological perspective, although it may still present an important advance in RNA biology.

Minor Points:

1. Fig. 2: The mRNA half-life of Cluster 1 does not appear to be significantly shorter than that of the other clusters. To clarify this, please compare the mRNA half-life of Cluster 1 with the average half-life across all clusters. It would be helpful to reference or extend this comparison using the data in Figure 2C.
2. Fig. 2D: Rather than comparing mRNA half-lives between Modules for each condition (noAct, Day 1, and Day 5), it would be more informative to show how the degradation rate changes with activation within each Module.
3. Fig. 2E, Line 269: "Indeed, the late-effector, memory precursor, and naïve/late-effector/memory genes (modules C, D, and F) contained more shorter-lived (<3 and 3-4 hours) than long-lived (4-6 and >6 hours) mRNAs, which were derived mainly from DESeq2 clusters 1 and 2." If you wish to support this claim effectively, it would be clearer to separate the coloring by modules A, C, D, and F. As it currently stands, the figure is difficult to interpret, making it challenging to evaluate the distribution of shorter- and longer-lived mRNAs.
4. Fig. 3: Please include a brief explanation in the text regarding why TNF α was chosen for analysis and its relevance to CD8 T cell activation. Providing context for the importance of TNF α in this specific process will help clarify the rationale behind its selection.
5. Fig. 3I: The values of GFP/Katushka for each condition should be shown as raw data rather than relative to the WT control. If you still wish to compare the data to WT, you can normalize by setting the mean of the WT values to 1 and plotting the other conditions accordingly.
6. Fig. 3J: How was the gating for GFP⁺ cells determined? Please clarify the criteria used for defining GFP positivity. Additionally, would it be more informative to compare the Mean Fluorescence Intensity (MFI) of GFP in Katushka-positive

cells, rather than focusing on percentage increases?

Reviewer #2

(Remarks to the Author)

The manuscript by Gameiro & Foskolou et al. describe the use of m6A profiling to explore the impact of both canonical and non-canonical m6A sites. Using miCLIP of m6A, they identify m6A sites within U-rich sequences that represent a novel subtype of m6A sites, and show that the combination of modification at both canonical and AU-rich sites may better explain the correlation between m6A modification and RNA stability. Overall, I think this finding is quite interesting, and the analysis of the m6A-CLIP is well done.

I think as a big-picture comment, Figures 1 and 2 are quite interesting and thorough; I'm a little less positive on the strengths of later sections. Figure 3 is interesting, but to me it's ultimately unclear whether it's a good representative example for the analysis in Figs 1-2 (particularly as deleting the RRACH motif seems to not alter RNA levels in Fig 3I). Similarly, the experiment in Fig 4a-c is quite interesting, but kind of unconnected to the rest of the manuscript – it seems like some additional data to connect this (does LSM knockdown preferentially affect stability of highly modified transcripts / binding overlap with YTH / etc) would be really helpful here to validate and connect the LSM finding, and having some further analysis of 4D would also be helpful to show that this isn't just an obvious effect of them being m6A-modified sites in Figure 1C/2A.

Minor comments:

- Is there evidence that indicates that the reverse transcription termination is antibody-driven and not due to the Adenosine methylation? I find it very surprising that an uncrosslinked antibody fragment (after proteinase K treatment) would stay bound strongly enough to make it through phenol:chloroform extraction and alter RT, whereas I'm not sure if anyone has shown that m6A modifications won't cause SSIV to read through or terminate at altered frequencies?

-

- As a related question, what nitrocellulose membranes are used? Some work has suggested that different versions from different suppliers show variable transfer of non-protein-crosslinked RNA (PMID 32252787), so it would be helpful to have that information here (I don't see it in the methods section)

- In figures like 1B / Sup 1B, I think it would make sense to label/group these into the RRACH/WWAWW groups that Fig 1C then goes on to describe them as – as I don't think it's easy to flip between Fig 1B and other figures to understand what's happening (I'm also not sure it's clear what Fig 1B has, as it's described in the legend as “the unique set of the 20 most enriched and variant 5-mers were plotted on the heatmap.”, but there seem to be 26 5-mers there? I'm also a bit confused in 1B/1C – it seems like in 1C the 'noUV' shows a peak at position 0, but in 1B UUAUU is notably depleted in no-UV?

- I'm not clear why the control in Fig 2B (/C/etc) is 'all clusters' and not 'genes lacking ARE/RRACH miCLIP peaks'? Having the comparison between cluster 2 and 3/4/5 is helpful, but it seems like comparing against non-m6A genes would also be a good control. (Also Fig 2B should be clear what the KS-test p-value is against – I assume it's 'all clusters' but I don't think it's clear?)

- It seems like the statistic in Fig 3E-F are comparing FTO against METTL3, but the text implies a comparison between control and FTO

- Fig 3G confuses me a little; it seems like each of the 3 replicates is normalized to both HPRT and then (not written) to the TNF level in control (which is why the values are all 1 in the control)? But then an unpaired t-test wouldn't be appropriate if the control values are all =1? It seems the more typical was to represent that (and 3E-F) would be to normalize the values by the average of the 3 control points, so the variability is still there

- In summary, it seems that the TNF example in Fig 3H-K goes against the 'having both causes an increased instability' – I get that the authors try to describe it as evidence of a more complex relationship (and the ARE there having a more dominant role than the RRACH one), but I think it would be good to connect this a bit more directly to the story presented in Figure 2

Version 1:

Reviewer comments:

Reviewer #1

(Remarks to the Author)

Thank you for carefully considering my comments. I appreciate the authors' efforts in revising the manuscript, which has clearly improved and now satisfactorily addresses all of my concerns.

Reviewer #2

(Remarks to the Author)

The authors have addressed my major concerns and I support publication

To clarify a prior comment – I agree with the response that the antibody fragment remaining after protK treatment in standard (UV-crosslinked) samples would cause termination; the question about reverse transcription termination was meant in reference to the no-UV samples in Fig. 1B (and the prior figure 1C, which more explicitly showed similar ‘crosslinks’ in no-UV samples). If I’m reading correctly, the PEKA analysis in that figure effectively quantifies single-nucleotide motif enrichment, but the no-UV samples also show enrichment for many of the RRACH and W-rich motifs (often at levels similar to the UV-crosslinked samples), which I find surprising?

As a minor comment that follows a prior comment – I appreciate the edits, the text for Fig1 now mainly refers to ‘miCLIP-ARE’ sites, but that terminology isn’t used anywhere in the figure (I think in the figure it’s the ‘WWAWW-miCLIP’ class) – it seems like it would make the text and figure more connected to include that naming scheme in the figure

Reviewer #3

(Remarks to the Author)

The manuscript by Gameiro et al is a study of how mRNA stability adjustments during CD8⁺ T-cell activation by the combined action of m⁶A methylation and AU-rich elements (AREs). Using miCLIP/iiCLIP and GLORI to map m⁶A, SLAM-seq to measure half-lives, and targeted assays (SELECT, reporter constructs, and RNA pulldown–proteomics), the study proposes a composite regulatory unit in which m⁶A at RRACH motifs adjacent to AREs helps recruit decay machinery (YTHDFs and LSMs), promoting rapid (“meta-unstable”) decay of key effector/memory transcripts. The work is timely and relevant for understanding post-transcriptional control in antitumor immunity.

The suggestions below are focused on to the mass spectrometry-related findings. Most are clarifications you can address in text, figure legends, or as supplement.

- In the first results section: ‘ARE-flanking m⁶A sites revealed by motif integration of miCLIP and GLORI profiling’, the consistently indicates that m⁶A sits on RRACH motifs that are flanked by nearby AREs. However, in the RNA pulldown assays the authors use a TNF 3’UTR and UUAUU probe where the m⁶A is placed on the ARE center while a downstream RRACH is left unmodified (except in the GGACU probe that the m⁶A modification is explicit mentioned). Those probe designs do not mirror the experiments established earlier (m⁶A on RRACH next to an ARE), so the pulldown enrichments are not directly extrapolable to the first situation. I recommend the authors state this explicitly and, if possible, test a complementary probe with m⁶A on the RRACH (ARE unmodified), or provide a brief rationale for using the ARE-centered methylation.

- Please add a short note on the rationale and limitations of using Jurkat instead of primary CD8⁺ cells

The MS/proteomics work is well designed and broadly supports the main biology findings. With the clarifications above especially where the m⁶A sits in the probes, the parts will be clear and convincing for general readers.

Reviewer #4

(Remarks to the Author)

REVIEWER COMMENTS

Reviewer #1 (Remarks to the Author):

Reviewer Comments:

This manuscript explores the correlation between N6-methyladenosine (m6A) and mRNA decay in human CD8 T cells using the miCLIP method. The authors report that the combined miCLIP signal for RRACH and ARE motifs is enriched in the 3'-UTR of meta-unstable mRNAs during CD8 T cell activation. They also investigate the roles of these motifs in TNF mRNA stability and claim that YTHDF1 binds to methylated AREs of meta-unstable mRNAs.

Strengths:

1. The authors constructed a valuable dataset using sophisticated techniques, offering a new classification of mRNAs in human CD8 T cells based on m6A modification sites and instability.
2. A variety of data analysis methods were applied, providing important insights into post-transcriptional gene regulation, particularly in the context of m6A and mRNA decay.

Weaknesses:

1. The study does not adequately explore the physiological impact of these motifs on CD8 T cell function. The functional relevance of the observed m6A modifications remains unclear.
2. The authors focus on TNF mRNA, which may not be the most critical for CD8 T cell function. A broader investigation into other mRNAs that are more central to CD8 T cell biology would strengthen the study's conclusions.

With the new insights from **GLORI transcriptomic data**, we now refer to miCLIP-ARE sites as “ARE-flanking m⁶A” sites.

We have now added data showing the effect of m⁶A on other targets that have ARE-flanking m⁶A sites. A prime example is *IL7R*, which is in cluster 2 of miCLIP dataset (Figure 2A) and cluster i of the newly generated GLORI dataset (Figure 2E) (Table S1 Fig S3D). Functionally, the inhibition of METTL3 increased *IL7R* mRNA stability, mRNA and protein in activated CD8 T cells levels (Figure 3A-F). *IL7Ra* has crucial roles in CD8 T cell homeostasis and function, especially for memory formation. During activation *IL7Ra*-high CD8 T cells are associated with memory precursors cells, whereas *IL7Ra*-low cells are associated with short-lived effector cells (PMID: 11062503, PMID: 18390712). This

is in line with results in Figure 2G, where we found that many meta-unstable genes are associated with T cell memory, including *IL7R*. In addition, we assessed the expression of other molecules associated with CD8 T cell activation and memory, such as CD69, CD28 and TCF7 upon METTL3 inhibition (Figure 3B). Although the donor-to-donor variation is high in primary CD8 T cells and these results did not reach significance, most donors showed increased mRNA levels for these targets upon METTL3 inhibitor treatment (Figure 3B).

In addition to the effects on mRNAs involved in CD8 T cell memory formation, we focused on genes associated with CD8 T cell function, namely the cytokines TNF- α and IFN γ . Among these, TNF- α is a key mRNA for functional assays in Figure 3, since a) has a high-confidence miCLIP-ARE site (Figure S3E), b) it is part of cluster 2 in miCLIP data (Figure 2A, Table S1), and c) it was previously shown to be regulated by m⁶A in CD4 T cells (PMID: 38657004). IFN γ has also been shown to be regulated by m⁶A in CD4 T cells (PMID: 34134995). Although IFN γ was found in cluster 4 of miCLIP data (Table S1), it contained a miCLIP-ARE site (Figure 3F, Table S1). Our results show that METTL3 inhibition increased the mRNA levels of *TNF*, *IFNG* and the chemokine *CCL4* (Figure 3B), whereas FTO inhibition decreases their protein expression (Figure 3H, 3I).

Our GFP reporter assays now also include the 3'UTR of IFN γ . FTO inhibition led to a decrease of GFP production when GFP is coupled with the 3'UTR of *IFNG* and *TNF*, but not when it was coupled with the 3'UTR of *Gzmb* – a T cell effector molecule with no high-confidence miCLIP-ARE site in its 3'UTR (Figure S3G).

The **new Figure 3** show additional evidence that ARE-flanking m⁶A motifs have functional significance in CD8 T cell memory formation and function. We hope these data enhance the immunological relevance of our study.

3. Consequently, while the findings are of interest within the field of RNA-binding proteins and post-transcriptional regulation, the immunological relevance is limited. This may reduce the manuscript's impact from an immunological perspective, although it may still present an important advance in RNA biology.

Minor Points:

1. Fig. 2: The mRNA half-life of Cluster 1 does not appear to be significantly shorter than that of the other clusters. To clarify this, please compare the mRNA half-life of Cluster 1 with the average half-life across all clusters. It would be helpful to reference or extend this comparison using the data in Figure 2C.

We revised Figure 2A to align with the analysis in Figure 1. Previously, Figure 2 included all significant crosslinks (PEKA-thresholded sites) overlapping W-rich motifs (WWWWW). We now restrict the analysis to PEKA sites overlapping WWAWW motifs centred on an adenosine (“A”), consistent with the approach in Figure 1C and S1B.

The updated miCLIP signatures in Figure 2 remain altered. Notably, the difference in half-lives between cluster 2 (enriched in miCLIP-ARE and miCLIP-RRACH sites) and “all genes” increased (Fig. 2B), likely due to the higher stringency applied when defining miCLIP crosslinks at ARE motifs.

Figures 2B and 2C display the same dataset but in different formats: Figure 2B shows the cumulative fraction of mRNA half-lives, whereas Figure 2C (Figure S2E in the resubmitted manuscript) presents the distribution of mRNA half-lives as bar plots, enabling comparison across T cell states.

In line with Reviewer 2’s suggestion, we have changed the reference group from “all clusters” to “all genes” in Figures 2B and S2E. To address the reviewer’s point further, we also introduced a new group - “mRNAs without detectable miCLIP signal (< 2 counts)” - labelled “genes.nomiCLIP”. We consider this a more appropriate reference, as the “all clusters” group is skewed by overrepresentation of both the shortest-lived mRNAs (cluster 2) and the longest-lived mRNAs (cluster 4).

2. Fig. 2D: Rather than comparing mRNA half-lives between Modules for each condition (noAct, Day 1, and Day 5), it would be more informative to show how the degradation rate changes with activation within each Module.

We agree with the reviewer. We show this information as Figure S2H.

3. Fig. 2E, Line 269: “Indeed, the late-effector, memory precursor, and naïve/late-effector/memory genes (modules C, D, and F) contained more shorter-lived (<3 and 3-4 hours) than long-lived (4-6 and >6 hours) mRNAs, which were derived mainly from DESeq2 clusters 1 and 2.” If you wish to support this claim effectively, it would be clearer to separate the coloring by modules A, C, D, and F. As it currently stands, the figure is difficult to interpret, making it challenging to evaluate the distribution of shorter- and longer-lived mRNAs.

We agree with the reviewer and have recoloured the modules in Figure 2G to improve contrast. We also removed the CD8 T cell state, showing only Day 5-activated T cells, which makes it easier to see that modules C, D, and F contain more short-lived than long-lived mRNAs, as indicated by the higher density of red and green lines in the lower

half of Figure 2G. Notably, these short-lived mRNAs connect more predominantly to DESeq2 cluster 2 and GLORI cluster i.

4. Fig. 3: Please include a brief explanation in the text regarding why TNF α was chosen for analysis and its relevance to CD8 T cell activation. Providing context for the importance of TNF α in this specific process will help clarify the rationale behind its selection.

We explain in results section the rationale for choosing TNF- α for targeted analysis: pp. 377-382 and pp. 408-410.

5. Fig. 3I: The values of GFP/Katushka for each condition should be shown as raw data rather than relative to the WT control. If you still wish to compare the data to WT, you can normalize by setting the mean of the WT values to 1 and plotting the other conditions accordingly.

New Figure 3M: The values are normalised to WT because we used the $2^{-\Delta\Delta C_t}$, i.e., all is compared to the WT condition. Katushka was used as a reference gene to account for transduction efficiency. However, we understand the importance to show the raw values also for the control, as the reviewer suggests. We have now included $2^{-\Delta C_t}$ values in Figure S3P, where the variability of the WT is also shown.

6. Fig. 3J: How was the gating for GFP $^+$ cells determined? Please clarify the criteria used for defining GFP positivity. Additionally, would it be more informative to compare the Mean Fluorescence Intensity (MFI) of GFP in Katushka-positive cells, rather than focusing on percentage increases?

New Figure 3P: GFP $^+$ gating was defined using the Katushka-only control, which lacks GFP-positive cells (Figure S3Q). As suggested by the reviewer, we now compare GFP mean fluorescence intensity (MFI), which is more informative (Figures 3O, 3P and Figure S3S, S3T)

Reviewer #2 (Remarks to the Author):

The manuscript by Gameiro & Foskolou et al. describe the use of m6A profiling to explore the impact of both canonical and non-canonical m6A sites. Using miCLIP of m6A, they identify m6A sites within U-rich sequences that represent a novel subtype of m6A sites, and show that the combination of modification at both canonical and AU-rich sites may better explain the correlation between m6A modification and RNA stability. Overall, I think this finding is quite interesting, and the analysis of the m6A-CLIP is well done.

I think as a big-picture comment, Figures 1 and 2 are quite interesting and thorough; I'm a little less positive on the strengths of later sections. Figure 3 is interesting, but to me it's ultimately unclear whether it's a good representative example for the analysis in Figs 1-2 (particularly as deleting the RRACH motif seems to not alter RNA levels in Fig 3I).

We have now tested more targets (IL7R, IFN γ , CD69, TCF7). IL7R is particularly a representative 3'UTR harbouring ARE-flanking m⁶A sites, since it a) comes up in all of our orthogonal datasets (miCLIP, GLORI and SLAM-seq), b) its 3'UTR harbours a high-confidence miCLIP-ARE site (Figure S3D), and c) it is a meta-unstable mRNA, with a half-life of 2.35 hr in noAct and 1.03 hr in Day5-activated CD8 T cell (Table S1).

New Figure 3: We observed increased stability, total mRNA levels, and protein expression for IL7R (Fig. 3A, 3E; Fig. S3A). Although we attempted to generate GFP constructs and mutants containing the IL7R 3'UTR, the large size of this region (3.1 kb) made cloning challenging. We therefore focused on TNF- α as an alternative, equally strong example of a 3'UTR harboring a high-confidence miCLIP-ARE site (Fig. S3E). In addition, we complemented the GFP reporter assays with constructs containing the 3'UTRs of IFN γ and GzmB (Fig. 3G)

Similarly, the experiment in Fig 4a-c is quite interesting, but kind of unconnected to the rest of the manuscript – it seems like some additional data to connect this (does LSM knockdown preferentially affect stability of highly modified transcripts / binding overlap with YTH / etc) would be really helpful here to validate and connect the LSM finding, and having some further analysis of 4D would also be helpful to show that this isn't just an obvious effect of them being m6A-modified sites in Figure 1C/2A.

We thank the reviewer for their suggestion regarding the role of LSM proteins, as revealed in Fig. 4A–C. We have performed preliminary experiments depleting LSM paralogues in primary CD8⁺ T cells, achieving only partial knockouts (Figure below). We

assessed the impact of LSM depletion on our target genes and observed substantial donor-to-donor variability, particularly for TNF- α , likely due to the incomplete knockouts and the inherent variability of primary T cell responses. Nonetheless, we detected a trend toward increased IFN γ expression following LSM6 depletion, and CXCR4 expression was also elevated.

These preliminary data suggest that LSM proteins may exert target-specific effects, with either increased or decreased expression depending on multiple factors, such as RBP competition or interactions with the RNA decay machinery. While we agree that this is an interesting aspect for future investigation, we consider it beyond the scope of the current study. We plan to explore the roles of these novel m⁶A-dependent RBPs in a dedicated follow-up manuscript.

Figure extra | Effect of LSM2 and LSM6 depletion on miCLIP-target mRNAs.

Primary CD8⁺ T cells were activated and, two days later, CRISPR/Cas9-edited with sgRNAs targeting LSM2 or LSM6. After 7 days, cells were assessed for TNF- α and IFN γ expression following 4 h reactivation with PMA/ionomycin, and for IL7R and CXCR4

expression without reactivation. **(A)** Western blot showing partial knockdown of LSM2 and LSM6 in two donors. **(B)** Raw MFI values for TNF- α , IFN γ , IL7R, and CXCR4 in four donors. **(C)** Corresponding MFI fold changes relative to control, based on the data in (B).

Minor comments:

- Is there evidence that indicates that the reverse transcription termination is antibody-driven and not due to the Adenosine methylation? I find it very surprising that an uncrosslinked antibody fragment (after proteinase K treatment) would stay bound strongly enough to make it through phenol:chloroform extraction and alter RT, whereas I'm not sure if anyone has shown that m⁶A modifications won't cause SSIV to read through or terminate at altered frequencies?

There is strong evidence that RT termination in CLIP experiments is primarily caused by the peptide remaining on RNA after crosslinking. Proteinase K digestion leaves a short peptide adduct, which can block reverse transcriptase and cause truncations or readthrough events. As noted by Sugimoto *et al.* (2012, PMID: 22863408): "*However, the peptide or amino acid left on the RNA after treatment with proteinase K can obstruct the reverse transcriptase, and therefore primer extension studies showed that a significant proportion of cDNAs truncate at the cross-link sites*" [Urlaub *et al.*, 2002, PMID: 12054894]. The effect is also influenced by the RT enzyme and buffer choice: different enzymes (SSIV, SSIII, TGIRT) produce variable readthrough and mutation (CIMS) frequencies at crosslink sites (PMID: 28790018). In our study, we used SSIV with an Mg²⁺-based buffer, consistent with the improved iiCLIP protocol (Lee *et al.*, 2021, Ref. 36).

To our knowledge, m⁶A does not disrupt Watson–Crick base pairing and thus should not, on its own, cause SSIV truncations or readthrough. In contrast, certain modifications such as m¹A impair base pairing and can stall reverse transcription, leading to truncations or mutations (PMID: 29908293). While systematic studies of m⁶A effects on RT fidelity are limited, current evidence does not support m⁶A alone as a direct cause of SSIV truncation or readthrough in the absence of crosslinked proteins. We thank the reviewer for this technical point.

- As a related question, what nitrocellulose membranes are used? Some work has suggested that different versions from different suppliers show variable transfer of non-protein-crosslinked RNA (PMID 32252787), so it would be helpful to have that information here (I don't see it in the methods section)

We thank the reviewer for raising this point, which we were not previously aware of. We used 0.45 µm nitrocellulose membranes (“Amersham Protran Premium,” Cytiva). Unfortunately, the supplier of “membrane G/I” in PMID: 32252787 is not specified, making direct comparison difficult.

Technical differences exist between our miCLIP protocol and the eCLIP protocol (PMID: 32252787) make it difficult to infer transfer efficiency of non-crosslinked RNA solely from the membrane type. The eCLIP method detects RNA via biotin labelling and chemiluminescence, whereas we used infrared (IR) imaging of an IR dye-conjugated adapter ligated to RNA 3' ends, as in the irCLIP protocol (PMID: 27111506). This approach is more specific than standard radiolabelling in iCLIP, as it reports only on RNA with successful 3' adapter ligation.

Fig. S1C shows the IR signal for all miCLIP samples and negative controls. For example, m⁶₂A, methylation is abundant in ribosomal RNA. We observed minimal IR signal above the 50 kDa antibody heavy-chain band in the anti-m⁶₂A control (3rd gel image on the right - lane 1), which indicates low nonspecific RNA transfer. By contrast, the strong IR signal observed for the non-crosslinked (no-UV) anti-m⁶A sample (lane 2), absent in both IgG and anti-m⁶₂A controls, supports the conclusion that the signal is m⁶A-specific and likely dependent on the antibody used.

- In figures like 1B / Sup 1B, I think it would make sense to label/group these into the RRACH/WWAWW groups that Fig 1C then goes on to describe them as – as I don't think it's easy to flip between Fig 1B and other figures to understand what's happening (I'm also not sure it's clear what Fig 1B has, as it's described in the legend as “the unique set of the 20 most enriched and variant 5-mers were plotted on the heatmap.”, but there seem to be 26 5-mers there? I'm also a bit confused in 1B/1C – it seems like in 1C the 'noUV' shows a peak at position 0, but in 1B UUAUU is notably depleted in no-UV?

We now present a single heatmap of enriched k-mers (Figure 1B).

Note: We incorporated an orthogonal m⁶A profiling using the GLORI method, which revealed that miCLIP-ARE sites correspond to AREs flanked by m⁶A within (±) 4 nucleotides of the A centre. We define this composite m⁶A motif as an ARE-flanking m⁶A site. Consequently, the miCLIP meta-distribution has been moved to Figure S1B, while the GLORI meta-profiles are shown in **new Figure 1C**.

Figure 1B shows motifs enriched at PEKA-thresholded sites, whereas Figures 1C and S1B align GLORI counts (Fig. 1C) and miCLIP crosslinks (Fig. S1B) to generic motif

classes. In Figure S1B, we also show miCLIP crosslinks at representative specific motifs (GGACU, UUAUU, and CGACU for RRACH, WWAWW, and CRACH, respectively).

To address the reviewer's concerns:

A) The legend "unique set of the 20 most enriched and variant 5-mers" refers to the union of the "20 most enriched" and "20 most variant" motif sets. The term "unique" reflects the R function used to remove duplicate entries from this union.

B) We have repeated the PEKA analysis using the most recent human genome annotation from flow.bio (Materials and Methods), which improves the mapping of repetitive regions - critical for accurate identification of PEKA-thresholded sites. The updated Fig. 1B reflects this analysis. The most enriched k-mers are consistent with our previous results, now also showing "UUAUU" enrichment in the noUV sample (third motif from the top).

C) Fig. S1B shows a UUAUU peak at position -1 in the noUV sample. Here, it is important to note that Fig. S1B plots the *number* of PEKA crosslinks (normalized by total crosslinks), whereas Fig. 1B plots the *scores* of PEKA-thresholded sites, which is a slightly more stringent metric (PMID: 36085079). Therefore, some variability in enriched k-mers can be expected between PEKA scores (Fig. 1B) and meta-alignments of PEKA counts (Fig. S1B).

- I'm not clear why the control in Fig 2B (/C/etc) is 'all clusters' and not 'genes lacking ARE/RRACH miCLIP peaks'? Having the comparison between cluster 2 and 3/4/5 is helpful, but it seems like comparing against non-m6A genes would also be a good control. (Also Fig 2B should be clear what the KS-test p-value is against - I assume it's 'all clusters' but I don't think it's clear?)

We agree with the reviewer that the 'all-clusters' group may be a biased representation of the shortest- and longest-lived mRNAs included in the DESeq2 model. **Figures 2B and S2E** now show the "all.genes" and "genes.nomiCLIP" as control groups.

- It seems like the statistic in Fig 3E-F are comparing FTO against METTL3, but the text implies a comparison between control and FTO

We thank the reviewer and have now fixed the text that refer to Figure 3L (pp. 421-423).

- Fig 3G confuses me a little; it seems like each of the 3 replicates is normalized to both HPRT and then (not written) to the TNF level in control (which is why the values are all 1 in the control)? But then an unpaired t-test wouldn't be appropriate if the

control values are all =1? It seems the more typical was to represent that (and 3E-F) would be to normalize the values by the average of the 3 control points, so the variability is still there.

New Figure S3C: We apologise for the earlier confusion. Our qPCR data were quantified using the $2^{-\Delta\Delta Ct}$ method. In this approach, each sample is first normalised to the housekeeping gene HPRT to account for differences in mRNA input levels ($2^{-\Delta Ct}$). The resulting values are then normalised to the control (CTL) condition to calculate the relative expression change ($2^{-\Delta\Delta Ct}$). This is a standard method for presenting qPCR results (PMID: 11846609).

We also thank the reviewer for noting that our original statistical test was inappropriate. An unpaired t-test on $2^{-\Delta\Delta Ct}$ values does not account for the pairing of samples across experimental conditions. We have now re-analysed the data using a paired t-test on the ΔCt values, which preserves the within-donor variability between the control and METTL3 KO conditions.

- In summary, it seems that the TNF example in Fig 3H-K goes against the ‘having both causes an increased instability’ – I get that the authors try to describe it as evidence of a more complex relationship (and the ARE there having a more dominant role than the RRACH one), but I think it would be good to connect this a bit more directly to the story presented in Figure 2.

We thank the reviewer for this accurate and insightful assessment. We agree that the epistatic relationship between ARE and RRACH motifs in regulating mRNA stability needs investigation beyond TNF mRNA, particularly in connection to the mRNA meta-instability described in Figure 2.

New Figures 3M–P: To further dissect the contributions of ARE and m⁶A motifs, we increased the number of donors and revised the quantification. The revised analysis confirms a trend towards decreased GFP mRNA levels in the ARE/RRACH double mutant compared to the ARE mutant alone ($P = 0.0546$), consistent with our original submission. Similarly, new Fig. 3P shows increased GFP protein levels (MFI) in the ARE mutant compared to WT and RRACH mutant conditions, but not compared to the double mutant.

We note that the RRACH mutant alone shows a trend towards lower transcript levels relative to WT=1 (Fig. 3M), indicating that an intact RRACH in the *TNF* 3'UTR stabilises mRNA - opposite to the canonical m⁶A decay effect. Together, these data suggest that mRNA instability is conferred specifically by m⁶A-flanking ARE sites in a RRACH-

dependent manner, and that the intact RRACH motif is required for the destabilising effect of interlinked AREs. The effect is stronger for m⁶A-flanking AREs than for m⁶A-modified RRACHs lacking nearby AREs.

We recognise that the effect sizes in Fig. 3M–P are modest. This is expected given (i) each construct differs by only a single nucleotide substitution (A>G), (ii) primary CD8⁺ T cells show substantial donor-to-donor variability, and (iii) ex vivo activation is required for these assays. Larger effects might be observed in transformed cell lines, but this would not capture the CD8⁺ T cell-specific m⁶A regulation we aimed to study. For example, GFP protein differences only emerged upon T cell activation (Fig. 3N–P).

In summary, these results support the model from Figure 2, in which RRACH-dependent mRNA decay is amplified when the m⁶A-modified RRACH is flanked by an ARE. According to this model (Figure 4E), the composite m⁶A motif acts as a key determinant of transcript meta-instability in activated CD8⁺ T cells, and the magnitude of decay can vary transcript-by-transcript depending on this motif context.

To:
Minju Ha, PhD
Senior Editor
Nature Communications

06th October 2025

Dear Dr. Ha:

We are pleased to submit our revised manuscript entitled “**Meta-unstable mRNAs in activated CD8+ T cells are defined by interlinked AU-rich elements and m6A mRNA methylation**” by Gameiro and Foskolou *et al.*

In this revised version, we have addressed all remaining minor comments by reviewers 2 and 3, as detailed below.

Reviewer #1 (Remarks to the Author):

Thank you for carefully considering my comments. I appreciate the authors’ efforts in revising the manuscript, which has clearly improved and now satisfactorily addresses all of my concerns.

We appreciate reviewer’s comments.

Reviewer #2 (Remarks to the Author):

The authors have addressed my major concerns and I support publication.

To clarify a prior comment – I agree with the response that the antibody fragment remaining after protK treatment in standard (UV-crosslinked) samples would cause termination; the question about reverse transcription termination was meant in reference to the no-UV samples in Fig. 1B (and the prior figure 1C, which more explicitly showed similar ‘crosslinks’ in no-UV samples). If I’m reading correctly, the PEKA analysis in that figure effectively quantifies single-nucleotide motif enrichment, but the no-UV samples also show enrichment for many of the RRACH and W-rich motifs (often at levels similar to the UV-crosslinked samples), which I find surprising?

We agree that m⁶A, by itself, does not disrupt Watson-Crick base pairing and therefore should not directly cause SSIV truncation or readthrough in the absence of crosslinking. This is consistent with the reviewer’s rationale and our original response.

To acknowledge this point, we highlight lines 144–155 about the surprising observation: that no-UV samples display an IR signal on the SDS–PAGE (Fig. S1C) and a corresponding truncation/deletion signature in cDNA (Figs. S1B, S1D). Our interpretation is that the anti-m⁶A antibody binds RNA with unusually high stability, surviving high-salt washes, SDS–PAGE separation, membrane transfer, and it appears that even after proteinase K digestion, antibody-derived peptides might still remain bound to the m⁶A site (lines 144–155). We thus propose that the IR signal and RT signatures in the no-UV condition arise from: (i) unusually stable anti-m⁶A:RNA binding resistant to stringent washes, and (ii) partial retention of antibody peptides on RNA after proteinase K digestion, which can block reverse transcription and produce truncation/deletion signatures. While unexpected, this interpretation best reconciles the observed IR signal and the sequencing patterns. That said, given that our study now includes GLORI data that is easier to interpret, this unusual behaviour of the antibody is not key to the conclusions we make.

We should add that our control experiments show that the strong IR signal is observed only in the anti-m⁶A no-UV samples, but not in IgG or when RNA is immunoprecipitated with an anti-m⁶₂A antibody, which recognises

the abundant m⁶₂A modification in ribosomal RNA. The latter shows only minimal nonspecific RNA transfer under noUV conditions (Fig. S1C, right panel, first lane), demonstrating that only the anti-m⁶A antibody has the capacity to remain bound to RNA in the absence of crosslinking.

As a minor comment that follows a prior comment – I appreciate the edits, the text for Fig1 now mainly refers to ‘miCLIP-ARE’ sites, but that terminology isn’t used anywhere in the figure (I think in the figure it’s the ‘WWAWW-miCLIP’ class) – it seems like it would make the text and figure more connected to include that naming scheme in the figure.

We thank the reviewer for pointing out the terminology inconsistency. As suggested, we have renamed the WWAWW sites in Figures 1D and 1E as miCLIP-ARE sites to match the text. Figure 1C and S1B were left unchanged, since it depicts the actual pentamer sequences centred at the miCLIP sites i.e. WWAWW, which correspond to the miCLIP-ARE sites. In addition, we revised lines 190-191 and the Figure 1 legend to clarify that miCLIP-ARE sites denote miCLIP crosslinks located at the centre of WWAWW pentamers. We believe these changes resolve the ambiguity and improve overall clarity.

Reviewer #3 (Remarks to the Author):

...

- In the first results section: ‘ARE-flanking m⁶A sites revealed by motif integration of miCLIP and GLORI profiling’, the consistently indicates that m⁶A sits on RRACH motifs that are flanked by nearby AREs. However, in the RNA pulldown assays the authors use a TNF 3’UTR and UUAUU probe where the m⁶A is placed on the ARE center while a downstream RRACH is left unmodified (except in the GGACU probe that the m⁶A modification is explicit mentioned). Those probe designs do not mirror the experiments established earlier (m⁶A on RRACH next to an ARE), so the pulldown enrichments are not directly extrapolable to the first situation. I recommend the authors state this explicitly and, if possible, test a complementary probe with m⁶A on the RRACH (ARE unmodified), or provide a brief rationale for using the ARE-centered methylation.

We thank the reviewer for raising this point and we clarify our rationale. In Figure 4, we compared (i) a canonical m⁶A-modified RRACH (Fig. 4a) and (ii) a composite probe where a RRACH (AGACU) is directly adjacent to an ARE (Fig. 4b), with m⁶A placed on the central adenosine of the ARE (AGACU-Um⁶AUU). We reasoned that AREs and nearby m⁶A sites act as a functional unit rather than independent elements.

We now clarify further in the text that this probe design was inspired by the original miCLIP signal, and it does not perfectly replicate the exact positioning of endogenous m⁶A sites as shown by GLORI. We designed the probe prior to obtaining GLORI data, but the probe nevertheless captures the composite architecture of RRACH + ARE sites to allow comparison with RRACH-only probes. Our observation that the custom ARE probe (Figure 4B) and TNF 3’UTR probe (Figure 4C) show the same enriched RBP families (YTH and LSM proteins), relative to the RRACH-only probe (Figure 4A), supports this conclusion. We now state this rationale explicitly in the Results (lines 478-482) to avoid overinterpretation of the pulldown data.

- Please add a short note on the rationale and limitations of using Jurkat instead of primary CD8⁺ cells.

We have clarified this rationale in the main text (lines 462-465) that: “Use of Jurkat T cells in these experiments provided sufficient material to reproducibly detect m⁶A ribonucleosides by LC-MS/MS, which could not be achieved with primary T cells due to lower RNA yield (see note in Materials and Methods).”

For further clarification, we added to the corresponding Materials and Methods (lines 1026-1032): “For CLIP-MS analysis of RBP-bound ribonucleosides, Jurkat T cells were UV-crosslinked, processed by CLIP, and the RNA was isolated and hydrolysed to ribonucleosides for LC-MS/MS analysis, as described below. The detected m⁶A ion counts were in the 0.5–5 fmol range, with signal-to-noise ratios of 15–30 relative to blanks. Although RBP abundance and regulation differ quantitatively between Jurkat and primary T cells, their

qualitative RNA-binding profiles are broadly conserved across cell types (given similar expression)⁸⁶, supporting the use of Jurkat T cells as a reductionist model for mechanistic studies of RBP binding and m⁶A methylation.”

The MS/proteomics work is well designed and broadly supports the main biology findings. With the clarifications above especially where the m6A sits in the probes, the parts will be clear and convincing for general readers.

Reviewer #4 (Remarks to the Author):

Thank you for your consideration. We look forward to the opportunity to share our work with the Nature Communications audience.

Sincerely,

Jernej Ule, Prof. of Neuroscience at King's College London

Randall S Johnson, Prof. of Molecular Physiology and Pathology,
University of Cambridge
